# Ideal Attribution and Faithful Watermarks for Language Models

**Min Jae Song** [1]   **Kameron Shahabi** [2]

## Abstract

We introduce ideal attribution mechanisms, a formal abstraction for reasoning about attribution decisions over strings. At the core of this abstraction lies the ledger, an append-only log of the prompt–response interaction history between a model and its user. Each mechanism produces deterministic decisions based on the ledger and an explicit selection criterion, making it well-suited to serve as a ground truth for attribution. We frame the design goal of watermarking schemes as faithful representation of ideal attribution mechanisms. This novel perspective brings conceptual clarity, replacing piecemeal probabilistic statements with a unified language for stating the guarantees of each scheme. It also enables precise reasoning about desiderata for future watermarking schemes, even when no current construction achieves them, as the ideal functionalities are specified first. In this way, the framework provides a roadmap that clarifies which guarantees are attainable in an idealized setting and worth pursuing in practice.

## 1. Introduction

Large language models (LLMs) are now routinely used for text generation, with applications ranging from composing emails to code generation and, to some degree, even mathematical proofs (Chatterji et al., 2025; Orabona & D'Orazio, 2025; Diez et al., 2026; Feldman & Karbasi, 2025). As LLMs improve in fluency, coherence, domain expertise, and reasoning capability, it becomes increasingly difficult to distinguish machine-generated text from human text. The scale of this proliferation is unprecedented; for the first time in history, large volumes of "human-grade" text are being produced by non-human processes.

How this shift will reshape the data ecosystem and its downstream products like LLMs, for better or worse, remains uncertain. What is clear, however, is that without provenance mechanisms in place, the shift is likely irreversible; once synthetic and human text are mixed without reliable markers of non-human origin, distinguishing between them may be nearly impossible (Köbis & Mossink, 2021; Clark et al., 2021; Jakesch et al., 2023). The development of provenance mechanisms is therefore urgent, not as a final solution, but as a necessary first step toward preserving the integrity of the current data ecosystem, providing time for the community to assess how it ought to be steered.

This work focuses on attribution and watermarking. Attribution concerns tracing text back to its origin, i.e., its "owner". Watermarking, a major focus of recent research, offers practical solutions to this attribution problem (Aaronson, 2022; Kirchenbauer et al., 2023a; Kuditipudi et al., 2023; Christ et al., 2024; Dathathri et al., 2024; Zhao et al., 2025). When the owner is an LLM service provider, attribution implies the ability to distinguish the provider's text from text produced by other generation processes. In this sense, attribution constitutes a refinement of the problem of distinguishing human and AI-generated text. For an LLM service provider, it subsumes the coarser human–AI distinction and imposes the stricter requirement of distinguishing its own text from that produced by other sources.

We add that attribution may also serve the business interests of LLM providers by substantiating ownership claims. For example, it can act as a mechanism to deter unauthorized model distillation, a major focus of recent disputes over DeepSeek (Metz, 2025; Mok, 2025). More broadly, attribution enables providers to monitor how their models are used in the wild, detect harmful use cases, and intervene when necessary. In these contexts, attribution mechanisms can provide evidence that complements the distinctive behavioral traits of deployed models.

**Ideal attribution mechanisms.** We introduce *ideal attribution mechanisms*, a formal abstraction for reasoning about attribution decisions over strings. Their main functionality is to answer questions such as, "given the full generation history of an LLM provider, is string $y$ attributable to that provider?" At the core of this abstraction lies the notion of a *ledger*, an append-only log of the prompt–response interac-

[1]Data Science Institute, University of Chicago, Chicago, IL, USA [2]Paul G. Allen School of Computer Science & Engineering, University of Washington, Seattle, WA, USA. Correspondence to: Min Jae Song <minjae.song@uchicago.edu>.

*Proceedings of the 43rd International Conference on Machine Learning*, Seoul, South Korea. PMLR 306, 2026. Copyright 2026 by the author(s).

tion history between a model and its user. Each mechanism outputs deterministic decisions based on the ledger and an explicit selection rule, making it well-suited to serve as a ground truth for attribution.

Our key novelty is to frame the design goal of watermarking schemes as faithful representation of ideal attribution mechanisms. In line with this framing, we formulate watermarking guarantees with respect to their ideal counterparts. This perspective brings conceptual clarity, replacing piecemeal probabilistic statements with a unified language for stating the guarantees of each scheme. It reduces confusion in interpretation and enables straightforward comparison of guarantees across schemes. For example, the semantics of a watermark detector's decisions, i.e., what it means for text to be watermarked under a given scheme, can be communicated directly by specifying the target ideal attribution mechanism. Because an ideal attribution mechanism is explicit and deterministic, it provides a complete specification of which responses are deemed attributable under any generation history, enabling transparent assessment of whether attribution decisions from the mechanism and its corresponding watermarking scheme are plausible.

Our framework departs from previous work on watermarking language models, which typically states guarantees for watermarking schemes without reference to a target ideal functionality, i.e., a ground truth (Aaronson, 2022; Kirchenbauer et al., 2023a; Kuditipudi et al., 2023; Christ et al., 2024; Dathathri et al., 2024; Fairoze et al., 2025). This practice has often led to ambiguity, misinterpretation, and fragmentation, with watermarking guarantees formulated differently across papers. One example is the subtle but important distinction between adaptive and non-adaptive robustness of watermarks, which went largely unnoticed until it was explicitly raised by Cohen et al. (2024). Another example is distortion-freeness, the guarantee that a watermarking scheme preserves the distribution of the original LLM, and thus its perceived quality. While single-query distortion-freeness can be ensured statistically (see, e.g., Kuditipudi et al. (2023)), such guarantees give a false sense of security over quality preservation since users typically interact with LLMs over multiple rounds. The more relevant guarantee is multi-query distortion-freeness, which is subsumed by the notion of watermark undetectability introduced by Christ et al. (2024).

**Watermarking schemes.** Watermarking is the practice of embedding a signal into content before its release so that the content can later be attributed to its origin. In this work we focus on schemes for language models that realize this by coupling generated responses with a random *watermark key*. Such schemes intervene during autoregressive token sampling, exploiting the fact that language models are randomized algorithms with explicit next-token probabilities.

Watermarking schemes aim to satisfy three main desiderata:

- **Distortion-freeness:** The output distribution of the watermarked model matches that of the original model, thereby preserving response quality.

- **Robustness:** The watermark remains detectable even after mild modifications to the watermarked output.

- **Soundness:** The watermark detector does not flag content that is not attributable to the watermarked model.

These high-level properties are often formalized in different ways. Distortion-freeness, for instance, has been formalized as watermark undetectability (Christ et al., 2024), i.e., computational indistinguishability between the watermarked model and the original model, $k$-query distortion-freeness (Kuditipudi et al., 2023), and heuristic checks based on manual inspection. Robustness likewise has been formalized in several ways, including non-adaptive robustness, where the perturbation adversary cannot observe previously sampled watermarked text; adaptive robustness (Cohen et al., 2024), where perturbations may depend on previous samples; and ideal security, where the adversary even has black-box access to the detector (Alrabiah et al., 2025). Definitions also vary with respect to the class of allowed perturbations, ranging from weak perturbations that leave sufficiently long uncorrupted substrings (Aaronson, 2022; Kirchenbauer et al., 2023a; Fairoze et al., 2025) to edit-distance perturbations (Kuditipudi et al., 2023; Golowich & Moitra, 2024).

For soundness, however, there is broad consensus, with prior work defaulting to the requirement that the detector not flag content generated *independently* of the watermark key. We view this formulation as too narrow since users cannot realistically be assumed to have zero contact with watermarked models in an LLM-saturated environment.

## 1.1. Main contributions

This work makes conceptual contributions to the study of attribution and watermarking for language models. Our core contribution is the introduction of ideal attribution mechanisms, an abstraction that provides a principled ground truth for watermarking schemes. Building on this foundation, we clarify desiderata for attribution, develop a unified language for expressing watermarking guarantees, and formalize properties not yet achieved by existing schemes, thereby charting a roadmap for future constructions.

**Formalization of ideal attribution (Section 2).** We define an ideal attribution mechanism as a sequence of time-indexed Boolean functions $(\mathcal{R}_t)$ over strings. For each time $t$, $\mathcal{R}_t : \{0,1\}^* \to \{0,1\}$ represents the ground-truth attribution decision with respect to the ledger $\Pi_t$, an append-only

record of all prompt–response interactions up to time $t$. In particular, $\mathcal{R}_t(\zeta) = 1$ indicates that the string $\zeta \in \{0,1\}^*$ is attributable to $\Pi_t$.

Each attribution function $\mathcal{R}_t$ is determined by $\Pi_t$ and a transcript-level attribution map

$$\mathsf{R} : \{0,1\}^* \times \{0,1\}^* \to 2^{\{0,1\}^*} \ ,$$

which is required to satisfy a set of axioms. A *transcript* consists of a single prompt–response pair $(x, u)$, and $\mathsf{R}(x, u)$ denotes the set of strings attributable to the response $u$ under prompt $x$.

This formalization enables transparent reasoning about ownership claims implicit in attributability (Section 2.2) and provides a principled language for expressing watermarking guarantees.

**Conceptual framework for watermarking (Section 3).** We frame watermarking schemes as *faithful* representations of ideal attribution mechanisms. Concretely, a watermarking scheme, specified by a sampler Wat and detector Ver, is faithful to an ideal attribution mechanism $\mathcal{R}$ if no probabilistic polynomial-time (PPT) adversary with black-box access to Wat and Ver can produce a string $\zeta$ for which $\mathsf{Ver}(\zeta) \neq \mathcal{R}(\zeta)$ (see Definition 3.1).[1] In words, faithfulness means that the verifier Ver essentially implements the ideal attribution function $\mathcal{R}$, without having to store the full conversation history.

This provides a unified language for expressing watermarking guarantees in terms of the target ideal attribution mechanism. When $\mathsf{R}$ is taken to be a robust attribution map (for some chosen notion of robustness), it yields a clean decoupling of the three main watermarking desiderata:

| | | |
|---|---|---|
| **Distortion-freeness** | $\equiv$ | Undetectability , |
| **Robustness** | $\equiv$ | Control over false negatives: |
| | | $\mathsf{Ver}(\zeta) < \mathcal{R}(\zeta)$ , |
| **Soundness** | $\equiv$ | Control over false positives: |
| | | $\mathsf{Ver}(\zeta) > \mathcal{R}(\zeta)$ . |

Control over false negatives corresponds to robustness; a scheme is adaptively robust if no PPT adversary $\mathcal{A}$, with black-box access to Wat and Ver, can produce a false-negative witness. Non-adaptive robustness corresponds to restricting $\mathcal{A}$ to a *single* sampling query to Wat.

Control over false positives yields a refined notion of soundness. In previous work, the default form of soundness corresponds to assuming independence from the watermark key.

---

[1]Strictly speaking, attribution is time-dependent. $\mathcal{R}$ denotes a sequence of ideal attribution functions indexed by the interaction time. We suppress this index for notational simplicity.

In our framework, this is equivalent to requiring false positive control only against adversaries with no prior interaction with Wat. By contrast, our framework allows soundness to be stated against adversaries with prior interaction.

To demonstrate that our definition of undetectable and faithful watermarking schemes is not vacuous, we reinterpret the security guarantees of ideal pseudorandom codes (PRCs) (Alrabiah et al., 2025) as faithfulness guarantees with respect to a specific ideal attribution mechanism under the uniform language model, i.e., language models that always assign equal probability to 0 and 1 (Section D). We then extend this interpretation to PRC-based watermarking schemes for general language models.

**Anticipating future watermarking schemes.** Our framework suggests a design paradigm, standard in cryptography: first specify the ideal functionality, and then design practical schemes to realize it, while making explicit how the achieved guarantees fall short of the ideal.

This ideal-first perspective enables reasoning about desiderata that no existing watermarking scheme achieves. As an illustrative case, we study unforgeability against adversaries with white box access to the detector. Prior work has considered only restricted forms of unforgeability, which are incompatible with the level of robustness required for useful watermarking. We sketch more general notions within our framework and conjecture that cryptographic primitives achieving these notions are technically feasible (see Section 3.2 and Section E).

Beyond its theoretical value, unforgeability could also serve the business interests of LLM service providers, since the presence of an unforgeable watermark constitutes strong, non-repudiable evidence of origin and would lend substantial weight to claims of model distillation.

## 2. Ideal Attribution Mechanism

A *ledger* $\Pi$ is the complete history of prompt-response interactions between a language model and its users. In principle, only text actually generated by the model, i.e., entries in the ledger, should be traceable to the model provider by any attribution mechanism. As such, our framework takes ledgers as the conceptual core for designing and analyzing watermarking schemes.

We start by defining ideal mechanisms for *verbatim* attribution, where only exact substrings of generated text are deemed attributable to the model provider. *Robust* attribution (Definition 2.7) is introduced later in Section 2.3. These mechanisms are *ideal* because they assume direct access to the ledger, which is itself an idealized object. Although unrealistic in practice, this serves as an uncontroversial gold standard for attribution accuracy and transparency. In Sec-

tion 2.2, we introduce *attribution soundness* as a property that endows attribution decisions with tenable semantics for exclusive ownership claims. In Section 2.4, we show how practical constraints in implementing ideal attribution mechanisms motivate the development of watermarking schemes.

**Language model.** A language model $Q : \{0,1\}^* \to [0,1]$ takes in a token sequence and outputs the probability that the next token is 1. Given a prompt $x \in \{0,1\}^*$, we obtain a *response* $u = (u_1, u_2, \ldots) \leftarrow Q(x)$ by autoregressive sampling initialized with $x$. For simplicity, we assume all responses have fixed length $\ell$, i.e., $u \in \{0,1\}^\ell$.

The transcript $\pi = (x, u)$ is a single prompt-response pair. Under autoregressive sampling from $x$, each bit $u_j$ of the response is sampled as

$$u_j \leftarrow \mathsf{Ber}\big(Q(xu_1 \cdots u_{j-1})\big) \,,$$

and we use the shorthand notation $u_j \leftarrow Q(\cdot \mid xu_{<j})$ or equivalently $u_j \leftarrow \bar{Q}(xu_{<j})$. We use $\bar{Q}_\ell(x)$ to denote the $\ell$-step autoregressive sampling oracle that, given a prompt $x$, samples a length-$\ell$ response $u \in \{0,1\}^\ell$ from $Q$.

**Ledger.** We formalize user interaction with a language model as a structured sequence of prompt-response sessions. Formally, the ledger $\Pi$ is a sequence of transcripts. That is,

$$\Pi = \big(\pi^{(1)}, \pi^{(2)}, \ldots\big) \,,$$

where $\pi^{(i)} = (x^{(i)}, u^{(i)})$ with $x^{(i)} \in \{0,1\}^*$ a user-provided prompt and $u^{(i)} \in \{0,1\}^\ell$ a fixed-length response by the language model.

**Ideal attribution mechanism.** Formally, an *ideal attribution mechanism* is given by a sequence of *attribution functions* $(\mathcal{R}_t)_{t \in \mathcal{T}}$ indexed by a *global time index* set

$$\mathcal{T} = \big\{(i,j) \mid i \in \mathbb{N} \,, j \in \{0,1,\ldots,\ell\}\big\} \,,$$

where time $t = (i,j)$ refers to the $j$-th response token $u_j$ in the $i$-th transcript $\pi^{(i)}$. The index $(i,0)$ refers to the prompt $x^{(i)}$. We order $\mathcal{T}$ lexicographically: $(i,j) < (i',j')$ if $i < i'$ or $i = i'$ and $j < j'$.[2]

At time $t = (i,j)$, the ledger $\Pi_t$ contains all completed transcripts together with the $j$-token prefix of the $i$-th transcript

$$\Pi_t = \big(\pi^{(1)}, \pi^{(2)}, \ldots, \pi^{(i-1)}, (x^{(i)}, u_1^{(i)}, \ldots, u_j^{(i)})\big) \,.$$

The *attribution function* $\mathcal{R}_t : \{0,1\}^* \to \{0,1\}$ depends on the ledger at time $t$ and a reference language model $Q$. For any string $\zeta \in \{0,1\}^*$, $\mathcal{R}_t(\zeta) = 1$ means that $\zeta$

---

[2]With fixed-length responses, the two-tuple index set $\mathcal{T}$ can be identified with $\mathbb{N}$ via a canonical bijection.

is deemed attributable to the ledger owner at time $t$, with the owner formally represented by the pair $(\Pi_t, Q)$. We also write $\mathcal{R}_t = \mathcal{R}^{\Pi_t}$ to emphasize dependence on the ledger. To make this notion more operational, we will later introduce *selection rules*, which provide an explicit criterion for attributability and a concrete representation of these attribution functions (Section 2.1).

## 2.1. Transcript-level attribution

Attributability should be *decidable at the moment a substring comes into existence*. Autoregressive generation can stop after any token, and the prefix observed up to that point is fixed and recorded in the ledger. Attribution for substrings of this prefix must therefore be determined immediately, without reference to future unseen tokens. Once made, an attribution decision is irrevocable; later tokens cannot retroactively change the status of substrings fully contained in the prefix, except if the same substring reappears as part of a longer continuation, in which case it may be reconsidered.

This motivates taking a single transcript as the natural scope of attribution decisions. Generation is autoregressive only within a transcript, so attribution decisions should depend only on the growing prefix of it. In addition, transcripts are structurally isolated within the ledger; substrings are defined only within a single transcript, and cross-transcript substrings do not exist. Hence, attribution decisions are made at the transcript level, and ledger-level attribution arises by aggregating these transcript-level decisions.

**String notation.** For $x, y \in \{0,1\}^*$, we write $x \sqsubseteq y$ to mean that $x$ is a prefix of $y$, and $xy$ to denote their concatenation. We use $\mathsf{substrings}(y)$, $\mathsf{prefixes}(y)$, and $\mathsf{suffixes}(y)$ to denote the sets of all substrings, prefixes, and suffixes of $y$. The symbol $\square$ denotes the empty string of length 0.

**Definition 2.1** (Transcript-level attribution)**.** *A transcript-level attribution map* $\mathsf{R} : \{0,1\}^* \times \{0,1\}^* \to 2^{\{0,1\}^*}$ *is a set-valued function satisfying the following axioms. For any prompt $x \in \{0,1\}^*$ and response $u \in \{0,1\}^*$, the set* $\mathsf{R}(x,u) \subseteq \mathsf{substrings}(u)$ *satisfies:*

1. *__Empty initialization.__ $\mathsf{R}(x,\square) = \emptyset$, where $\square$ denotes the empty string.*

2. *__Monotonicity in transcript history.__ If $u' \sqsubseteq u$ (i.e., is a prefix of), then*

$$\mathsf{R}(x,u') \subseteq \mathsf{R}(x,u) \,.$$

   *That is, extending the transcript cannot remove previously included strings.*

3. *__Suffix-jump property.__ For $u = u_1 \cdots u_j$,*

$$\mathsf{R}(x, u_{1:j}) \setminus \mathsf{R}(x, u_{1:j-1}) \subseteq \mathsf{suffixes}(u_{1:j}) \,.$$

*That is, new attributions can only arise as response suffixes ending in the latest token $u_j$.*

We say that a string $\zeta \in \{0, 1\}^*$ is *attributable* to a transcript $\pi = (x, u)$ if $\zeta$ lies in $\mathsf{R}(x, u)$. We define the associated attribution function $\mathcal{R}^\pi : \{0, 1\}^* \to \{0, 1\}$ by

$$\mathcal{R}^\pi(\zeta) = 1 \quad \Longleftrightarrow \quad \zeta \in \mathsf{R}(x, u) \ .$$

We now lift transcript-level decisions to the ledger. The ledger-level decision is simply the max (equivalently, logical OR) of transcript-level decisions.

**Definition 2.2** (Ledger-level attribution)**.** *Given a transcript-level attribution map $\mathsf{R}$ (Definition 2.1) and a ledger $\Pi = (\pi^{(1)}, \ldots, \pi^{(m)})$ with $\pi^{(i)} = (x^{(i)}, u^{(i)})$, we define the set of attributable strings with respect to $\Pi$ and $\mathsf{R}$ by*

$$\mathsf{R}(\Pi) = \bigcup_{i=1}^m \mathsf{R}\big(\pi^{(i)}\big) \ .$$

We write $\mathcal{R}_t = \mathcal{R}^{\Pi_t}$ for the attribution function at global time $t$, which is defined by

$$\mathcal{R}_t(\zeta) = 1 \quad \Longleftrightarrow \quad \zeta \in \mathsf{R}(\Pi_t) \ .$$

Each axiom in Definition 2.1 is straightforward to motivate on its own, but together they specify the mechanism only at an abstract level. For a more direct view, we introduce *selection rules*, which represent attribution decisions made on the spot as substrings arise.

**Selection rules.** A *selection rule* $\mathsf{Z}$ maps a triple $(x, \rho, \zeta)$, consisting of a prompt $x$, a response prefix $\rho$, and a response suffix $\zeta$, to an attribution decision in $\{0, 1\}$. For each transcript, $\mathsf{Z}$ induces a sequence of selection *vectors* $(z_j^{(i)})_{j \in [\ell]}$ naturally paired with the response tokens. Specifically, we define for any $i \in \mathbb{N}$ and $j \in \{1, \ldots, \ell\}$,

$$z_j^{(i)} = \big(z_{1,j}^{(i)}, z_{2,j}^{(i)}, \ldots, z_{\ell,j}^{(i)}\big) \in \{0, 1\}^\ell \ ,$$
$$\text{where} \quad z_{k,j}^{(i)} = \begin{cases} \mathsf{Z}(x^{(i)}, u_{1:k}^{(i)}, u_{k:j}^{(i)}) & \text{if } k \le j \ , \\ 0 & \text{else} \ . \end{cases}$$

The natural pairing of response tokens and selection vectors is then

$$\pi^{(i)} = \big(x^{(i)}, (u_1^{(i)}, z_1^{(i)}), (u_2^{(i)}, z_2^{(i)}), \ldots, (u_\ell^{(i)}, z_\ell^{(i)})\big) \ .$$

We say that a string $\zeta$ is *attributable* to a transcript $\pi = (x, (u_1, z_1), \ldots, (u_\ell, z_\ell))$ paired with selection vectors of $\mathsf{Z}$ if there exist $k \le j$ such that $u_{k:j} = \zeta$ and $z_{k,j} = 1$. Trivial examples of selection rules include the constant rule $\mathsf{Z} = 1$ (i.e., every substring of generated response is

deemed attributable) and $\mathsf{Z} = 0$ (i.e., no string is deemed attributable).

Proposition B.1 shows that any transcript-level attribution map can be represented by a selection rule, and vice versa. This provides formal justification for working with selection rules going forward. We prefer this view because selection rules are more direct and interpretable.

## 2.2. Meaningful attribution via soundness

We view attribution as the model provider's ownership claim over responses recorded in the model's ledger. The weight of an ownership claim rests on *exclusivity*: membership in the attribution set $\mathsf{R}(\Pi)$ should not be automatic or easily guaranteed. It should convey more than mere appearance in the ledger, meaningfully separating its members from any strings that have appeared or could arise independently in the world.

For instance, copy-instruction prompts such as "copy: ..." can cause language models to reproduce arbitrary strings with near certainty, allowing an adversary to push arbitrary strings into the ledger through simple prompting. If responses generated in this way are not rejected by the selection rule, then any string chosen in advance, for example an entire page of *The Hobbit*, could be planted into the attribution set. An attribution mechanism that accepts such on-demand manufactured responses cannot support a convincing notion of ownership.

We impose a soundness condition that requires the selection rule to reject responses that dilute exclusivity. This ensures that membership in the attribution set carries evidential value as an ownership claim. Formally, soundness requires that strings fixed in advance of ledger generation do not appear in the attribution set with overwhelming probability, even if a prompting adversary attempts to plant them.

Here and throughout, $\mathrm{negl}(\lambda)$ denotes a negligible function of $\lambda$, i.e., one that decreases faster than any inverse polynomial.

**Definition 2.3** (Attribution soundness)**.** *Let $Q : \{0, 1\}^* \to [0, 1]$ be a language model, and let $\lambda \in \mathbb{N}$ be the security parameter.[3] We say a sequence of attribution mechanisms $(\mathsf{R}_\lambda)_{\lambda \in \mathbb{N}}$ (Definition 2.2) is* sound *with respect to $Q$ if for any fixed $y \in \{0, 1\}^*$, any $\ell, m = \mathrm{poly}(\lambda)$, and any adversary $\mathcal{B}$ restricted to at most $m$ sampling queries,*

$$\Pr\left[\Pi \leftarrow \mathcal{B}^{\bar{Q}}(1^\lambda, y) : \mathcal{R}_\lambda^\Pi(y) = 1\right] \le \mathrm{negl}(\lambda) \ ,$$

*where $\Pi$ is the ledger generated by the interaction between*

---

[3]The *security parameter* $\lambda$ is a scaling knob that sets the complexity of cryptographic schemes (e.g., key sizes), and ensures that any attack running in time polynomial in $\lambda$ succeeds with only negligible probability.

$\mathcal{B}$ *and the sampling oracle* $\bar{Q}$, *which returns a length-$\ell$ string in response to each prompt of* $\mathcal{B}$.

**Interpretation of attribution soundness.** If an attribution mechanism is sound, then any string fixed before the realization of the ledger is extremely unlikely appear in the attribution set. In particular, common phrases that are widely regarded as unattributable to any specific entity will practically never appear in the attribution set since the set of all common phrases can be specified prior to the ledger-generating interaction.

Furthermore, a positive attribution decision $\mathcal{R}(y) = 1$ means that $y$ is uniquely assigned to the ledger owner in the sense that it belongs *only* to $\mathsf{R}(\Pi)$ and not to the attribution set of any independently generated ledger $\Pi'$. If two distinct model providers were to deploy the same language model $Q$, their respective attribution sets $\mathsf{R}(\Pi)$ and $\mathsf{R}(\Pi')$ would be disjoint except with negligible probability. This uniqueness embodies the clearest form of exclusivity and serves as the basis for interpreting attribution as ownership.

We provide sufficient and necessary conditions for selection rules to satisfy attribution soundness in Section B. Informally, given a prompt $x \in \{0,1\}^*$, a response prefix $\rho \in \{0,1\}^*$, and a response suffix $y \in \{0,1\}^*$, the selection rule $\mathsf{Z}$ should rarely select $y$ if $y$ is likely to be sampled under $\bar{Q}(\cdot \mid x\rho)$. For instance, with the copy-instruction prompt $x =$ "copy: $y$" and $\rho = \square$ (empty string), marking $y$ as an attributable response would clearly violate this condition. A trivial but vacuous selection rule satisfying soundness is one that rejects all responses, i.e., $\mathsf{Z} \equiv 0$.

### 2.3. Predicate-based expansion and robust attribution

We use predicates to formalize "closeness" between strings. From these predicates, we define expansions of individual strings, and then extend the definition to sets of strings.

**Definition 2.4** (Predicate). *A predicate is a Boolean-valued function* $\Phi : \mathcal{X} \to \{0,1\}$ *defined on some domain* $\mathcal{X}$.

In this work, we fix the domain to be $\mathcal{X} = \cup_{n \in \mathbb{N}}\big(\{0,1\}^n \times \{0,1\}^n\big)$, so that $\Phi$ takes as input a pair of equal-length binary strings and outputs a bit. A canonical example is the Hamming predicate.

**Definition 2.5** (Hamming). *For any* $n \in \mathbb{N}$ *and* $r = r(n)$, *we define the Hamming predicate* $\mathsf{Ham}_r$ *as follows. For any equal length strings* $y, y' \in \{0,1\}^n$

$$\mathsf{Ham}_r(y, y') = (\|y - y'\|_0 \leq r)$$

**Definition 2.6** (Predicate-based expansion). *Let* $\Phi : \mathcal{X} \to \{0,1\}$ *be a predicate. For any* $y \in \{0,1\}^*$, *its* $\Phi$-*expansion is defined by*

$$\Phi(y) = \{\zeta \in \{0,1\}^* \mid \Phi(y, \zeta) = 1\} .$$

*For any set* $A \subseteq \{0,1\}^*$, *its* $\Phi$-*expansion is defined by*

$$\Phi(A) = \cup_{y \in A} \Phi(y) .$$

*Given two predicates* $\Phi_1, \Phi_2$, *the expansion of their composition* $\Phi_2 \circ \Phi_1$ *is defined by*

$$\Phi_2 \circ \Phi_1(y) = \Phi_2(\Phi_1(y)) .$$

In particular, for an attribution set $\mathsf{R}(\Pi)$ we simply denote its $\Phi$-expansion by $\Phi(\mathsf{R}(\Pi))$, the set obtained by applying $\Phi$ to every string in $\mathsf{R}(\Pi)$. Its membership function is written as $\mathcal{R}^{\Phi,\Pi} : \{0,1\}^* \to \{0,1\}$.

**Definition 2.7** (Robust attribution). *Let* $\Phi : \mathcal{X} \to \{0,1\}$ *be any predicate, and let* $\mathsf{R} : \{0,1\}^* \times \{0,1\}^* \to 2^{\{0,1\}^*}$ *be any transcript-level attribution map (Definition 2.1). Then, For any transcript* $\pi \in \{0,1\}^* \times \{0,1\}^*$, *the* $\Phi$-*robust attribution set with respect to* $\mathsf{R}$ *is defined as* $\Phi(\mathsf{R}(\pi))$, *and denote the corresponding attribution function by* $\mathcal{R}^{\Phi,\pi}$

### 2.4. From ideal attribution to watermarking

Ideal attribution mechanisms serve as a clean conceptual model, but are unsuitable for real-world use. They rely on the assumption of direct access to the ledger, an assumption that is undesirable for practical deployment for several reasons such as:

1. **Privacy liability.** Model providers risk liability if transcripts containing private user data are leaked. Explicit ledger storage also creates a significant compliance burden under consumer data protection regulations such as the European Union's General Data Protection Regulation (GDPR).

2. **Service design.** Some providers, such as ChatGPT Enterprise (OpenAI, 2025), provide "zero data retention" as a privacy feature, meaning they do not explicitly store user prompts or responses.

3. **Operational overhead.** If *publicly-verifiable* attribution is desired, the ledger would need to be continuously updated, shared in real time, and accessed in full whenever an attribution query is made.

Our goal in watermarking is to retain the clarity of ideal attribution decisions while avoiding explicit storage of the ledger or any representation of it that requires continual updates. Concretely, this means selecting a target ideal attribution mechanism and designing a watermarking scheme that computes its attribution decisions without direct access to the ledger. This stands in contrast to prior watermarking schemes (Kuditipudi et al., 2023; Christ et al., 2024; Christ & Gunn, 2024; Fairoze et al., 2025), which lack "ideal" mechanisms whose functionality they are designed

to preserve. A notable exception is the work on ideal pseudo-random codes (Alrabiah et al., 2025), which can be viewed as watermarking schemes for the uniform language model.

A key feature of our approach is the conceptual clarity it inherits from the ideal attribution mechanism. When a verifier, operating without access to the ledger, outputs a decision bit on a given string, faithfulness of the watermarking scheme ensures that this decision is essentially identical to that of the ideal attribution mechanism. The ideal mechanism then provides a precise interpretation of the decision, explaining *why* the string was marked as attributable or not.

## 3. Undetectable and Faithful Watermarking

Watermarking schemes realize ideal attribution mechanisms without explicitly maintaining the ledger. Instead, the ledger is replaced by a succinct *watermark key* fixed at the start of the ledger-generating process. A watermark key $\mathsf{wk} = (\mathsf{pk}, \mathsf{sk})$ consists of a public component $\mathsf{pk}$ and a secret component $\mathsf{sk}$. Technically speaking, our faithfulness definition is formulated in a secret-key setting, since no part of the watermark key is revealed to a black-box adversary. Nevertheless, we decompose the watermark key into public and secret components to naturally extend the definition to publicly verifiable watermarking schemes, where adversaries have direct access to $\mathsf{pk}$.

Below we formally define undetectable and faithful watermarking schemes, and provide intuitive interpretations of the technical statements. We address the apparent type mismatch between the time-invariant watermark key and the time-evolving ledger in Section 3.1.

In Section D, we show that the definition is non-vacuous by demonstrating that ideal PRCs can be viewed as undetectable and faithful watermarking schemes for a canonical attribution map and the uniform language model $\mathcal{U} \equiv 1/2$. We then study the guarantees achieved by PRC-based watermarking schemes for general language models.

**Definition 3.1** (Undetectable and faithful watermarking). *Let $Q : \{0, 1\}^* \to [0, 1]$ be a language model, let $\Phi$ be an efficiently computable predicate, and let $\mathsf{R} = (\mathsf{R}_\lambda)_{\lambda \in \mathbb{N}}$ be an efficiently computable transcript-level attribution map. An* undetectable *and* faithful *watermarking scheme for $Q$ and $\mathsf{R}^\Phi$ is a triple of PPT algorithms* (Gen, Wat, Ver) *satisfying the following properties.*

**Undetectability:** *For any PPT adversary $\mathcal{A}$,*

$$\left| \Pr[\mathcal{A}^{\bar{Q}}(1^\lambda) = 1] - \Pr[\mathcal{A}^{\mathsf{Wat_{sk}}}(1^\lambda) = 1] \right| \le \mathrm{negl}(\lambda) \,.$$

**Faithfulness:** *For any PPT adversary $\mathcal{A}$, let* $(\mathsf{pk}, \mathsf{sk}) \leftarrow$ Gen$(1^\lambda)$, *and let* $\zeta \leftarrow \mathcal{A}^{\mathsf{Wat_{sk}}, \mathsf{Ver_{pk}}}(1^\lambda)$ *denote the string output by $\mathcal{A}$ at time $s \in \mathcal{T}$ ($\mathcal{A}$ may continue*

*interacting with the oracles thereafter). Then,*

$$\Pr\left[\mathsf{Ver_{pk}}(\zeta) \ne \mathcal{R}^\Phi_{\lambda,s}(\zeta)\right] \le \mathrm{negl}(\lambda) \,,$$

*where $\mathcal{R}^\Phi_{\lambda,s}$ denotes the $\Phi$-expansion of the attribution function $\mathcal{R}_{\lambda,s}$, determined by the ledger $\Pi_s$ and the attribution map $\mathsf{R}_\lambda$.*

For notational convenience, we henceforth suppress the security parameter $\lambda$ when clear from context (e.g., writing $\mathcal{R}^\Phi_t$ instead of $\mathcal{R}^\Phi_{\lambda,t}$).

**Remark 3.2** (Efficient computability). *Because the language model $Q$, predicate $\Phi$, and transcript-level attribution map $\mathsf{R}$ are all efficiently computable, a PPT adversary can simulate them directly. Hence, undetectability and faithfulness must hold even against adversaries with access to $Q$ and the entire sequence of attribution functions $(\mathcal{R}^\Phi_t)_{t \in \mathcal{T}}$.*

**Interpretation of undetectability.** Undetectability requires that no efficient adversary can distinguish between the sampling oracle $\bar{Q}$ and the watermarked oracle $\mathsf{Wat_{sk}}$, even with multiple interactions and access to next-token probabilities from $Q$. This represents the strongest form of quality preservation: the watermarked model generates transcripts of essentially the same *quality* as those of the unwatermarked model.

**Interpretation of faithfulness.** Faithfulness requires that $\mathsf{Ver_{pk}}(\zeta)$ equal the attribution function $\mathcal{R}^\Phi_s(\zeta)$, where $s$ is the time at which the adversary outputs the candidate string $\zeta$. The adversary has black-box access to the oracles $\mathsf{Wat_{sk}}$ and $\mathsf{Ver_{pk}}$, and we interpret such adversaries as modeling *honest* users. That is, they use the public key $\mathsf{pk}$ solely for its intended purpose of verification through $\mathsf{Ver_{pk}}(\cdot)$, without attempting to exploit it. Thus, for honest users, querying $\mathsf{Ver_{pk}}$ at time $s$ is effectively equivalent to querying the ideal attribution function $\mathcal{R}^\Phi_s$, which in principle requires explicit maintenance of the ledger.

The failure event in faithfulness is $\mathsf{Ver_{pk}}(\zeta) \ne \mathcal{R}^\Phi_s(\zeta)$, which splits into two events: false positives $\mathsf{Ver_{pk}}(\zeta) > \mathcal{R}^\Phi_s(\zeta)$, and false negatives $\mathsf{Ver_{pk}}(\zeta) < \mathcal{R}^\Phi_s(\zeta)$. A false positive means that $\mathsf{Ver_{pk}}$ accepts a string $\zeta$ not actually attributable under $\mathcal{R}^\Phi_s$, i.e., one that is $\Phi$-far from every string in $\mathsf{R}(\Pi_s)$. Conversely, a false negative means that $\mathsf{Ver_{pk}}$ rejects a string that is in fact attributable under $\mathcal{R}^\Phi_s$, i.e., one that is $\Phi$-close to some string in the attribution set $\mathsf{R}(\Pi_s)$.

**Stronger soundness for honest LLM users.** In Section 3.1, we introduce a stronger soundness property than Definition 2.3, which ensures that $\mathcal{R}^\Phi_s(\zeta) = \mathcal{R}^\Phi_T(\zeta)$, where $T$ is the termination time of the ledger. This means that the attribution decision for $\zeta$ is forward-stable. That is, once $\zeta$ is queried for attribution at its generation time $s$,

the decision will, with overwhelming probability, remain unchanged when the ledger is fully extended to $T$.

For honest LLM users, this means that if one ensures $\zeta$ is not robustly attributable to the responses observed so far, then $\zeta$ will, with overwhelming probability, remain unattributed as the ledger continues to grow in the future. Thus, even without querying $\mathsf{Ver}_{\mathsf{pk}}$, the user can be confident that a string $\zeta$ generated in this way will not later be flagged by $\mathsf{Ver}_{\mathsf{pk}}$. This yields a strengthened soundness guarantee for honest users, since it continues to hold even when the user has prior interaction with $\mathsf{Wat}_{\mathsf{sk}}$ before generating $\zeta$.

**Replacing the ledger with a fixed key.** Watermarking schemes achieve their guarantees by coupling the responses generated by $\mathsf{Wat}_{\mathsf{sk}}$ with the public key $\mathsf{pk}$. The built-in dependence structure enables computation of attribution decisions without access to the ledger.

In effect, the public key $\mathsf{pk}$, generated and announced at genesis, serves as a succinct commitment to the full ledger that unfolds over time. Committing at time zero resolves the issues with ideal attribution mechanisms, previously discussed in Section 2.4.

1. **Privacy of transcripts.** The watermark key $(\mathsf{pk}, \mathsf{sk})$ is sampled and fixed before any interaction with $\mathsf{Wat}_{\mathsf{sk}}$. Since this occurs prior to receiving user-provided prompts, and the scheme does not store any part of the growing ledger, transcript privacy is preserved.

2. **Low communication overhead.** Fixing the detection key $\mathsf{pk}$ at the start significantly reduces communication overhead relative to ideal attribution, since $\mathsf{pk}$ is succinct and does not require continuous updates.

### 3.1. Reconciling time-invariant verification and time-dependent attribution

The faithfulness guarantee $\mathsf{Ver}_{\mathsf{pk}}(\zeta) = \mathcal{R}_s^\Phi(\zeta)$ is tied to the specific time $s$ at which $\mathsf{Ver}_{\mathsf{pk}}$ is queried with $\zeta$. While $\mathsf{Ver}_{\mathsf{pk}}$ is time-invariant, the ideal attribution mechanism is query time dependent, represented by a sequence of attribution functions $(\mathcal{R}_t)_{t \in \mathcal{T}}$. This raises the question: given a string $\zeta$ output at time $s$, what can be said about the relation between $\mathcal{R}_s(\zeta)$ and $\mathcal{R}_t(\zeta)$ for $s < t$, beyond trivial monotonicity $\mathcal{R}_s(\zeta) \le \mathcal{R}_t(\zeta)$? In particular, can we guarantee that the undesirable event $\mathcal{R}_s(\zeta) < \mathcal{R}_t(\zeta)$ occurs only with negligible probability?

We define a stronger soundness property, called *anytime soundness*[4], which formalizes forward stability of ideal attribution functions (Definition 3.4). In Section C, we show

---

[4] The term "anytime" is inspired by the notion of anytime validity in sequential testing (Ramdas et al., 2023).

that anytime soundness is in fact *necessary* for the existence of a faithful watermarking scheme targeting it.

For an ideal attribution mechanism to satisfy this property, however, it is necessary to coarsen time to a strict subset $\mathcal{T}' \subset \mathcal{T}$. Without coarsening, an adversary observing the growing transcript can choose an "edge-of-inclusion" string $\zeta$ that, depending on the next sampled bit, has a high probability of entering the attribution set, and then output a guess of its completion (see Example C.1).

**Definition 3.3** (Time-aligned adversaries). *Let $\bar{Q}$ be a sampling oracle and let $\mathcal{T}' \subseteq \mathcal{T}$. An interaction adversary $\mathcal{B}$ is called* time-aligned *with $\mathcal{T}'$ if, in any execution, it chooses an* output time $s \in \mathcal{T}'$, *at which it produces $\zeta \in \{0,1\}^*$, and a* termination time $t \in \mathcal{T}'$ *at which it halts, with $s \le t$. The times $s$ and $t$ may be randomized and may depend on the preceding interaction history (i.e., they are stopping times), but must always lie in $\mathcal{T}'$.*

**Definition 3.4** (Anytime soundness). *Let $Q : \{0,1\}^* \to [0,1]$ be a language model, let $\Phi : \mathcal{X} \to \{0,1\}$ be a predicate, and let $\lambda \in \mathbb{N}$ be the security parameter. A sequence of attribution mechanisms $(\mathsf{R}_\lambda)_{\lambda \in \mathbb{N}}$ is* anytime sound *with respect to model $Q$, predicate $\Phi$, and coarsened time set $\mathcal{T}'_\lambda \subset \mathcal{T}$ if the following holds.*

*For any $\ell, q = \mathrm{poly}(\lambda)$ and any time-aligned adversary $\mathcal{B}$ with respect to $\mathcal{T}'_\lambda$ making at most $q$ sampling queries to $\bar{Q}$, where $\bar{Q}$ is an autoregressive sampling oracle for length-$\ell$ responses from $Q$,*

$$\Pr\left[\zeta \leftarrow \mathcal{B}^{\bar{Q}}(1^\lambda) : \mathcal{R}_{\lambda,s}^\Phi(\zeta) < \mathcal{R}_{\lambda,T}^\Phi(\zeta)\right] \le \mathrm{negl}(\lambda),$$

*where $\zeta \in \{0,1\}^*$ is the (early) output of $\mathcal{B}$, $s \in \mathcal{T}'_\lambda$ is the time at which $\zeta$ is produced, and $T$ is the termination time of the interaction between $\mathcal{B}$ and $\bar{Q}$.*

**Interpretation of anytime soundness.** Anytime soundness means that, while $\mathsf{R}(\Pi_T) \setminus \mathsf{R}(\Pi_s)$ may be nonempty, it is statistically hard to predict, at time $s \in \mathcal{T}'$, which specific strings (if any) will appear in this set difference. In contrast, basic soundness in Definition 2.3 corresponds to the special case $s = (0,0)$, meaning that it is statistically hard to predict, at time zero, which strings will appear in $\mathsf{R}(\Pi_T)$.

### 3.2. Beyond faithfulness

Faithfulness may be unattainable for certain target attribution mechanisms, even against black-box adversaries (see Example D.12 and Lemma D.14). In such cases, we can proceed nonetheless by *sandwiching* the semantics of the verifier $\mathsf{Ver}$ between two ideal attribution mechanisms. Specifically, we choose a pair of transcript-level attribution maps $(\mathsf{R}, \mathsf{S})$ satisfying $\mathsf{R}(x, y) \subseteq \mathsf{S}(x, y)$ for all prompts $x$ and responses $y$. The goal is to guarantee that

$$\mathcal{R}_s(\zeta) \le \mathsf{Ver}(\zeta) \le \mathcal{S}_s(\zeta),$$

where $s$ denotes the query time for $\mathsf{Ver}(\zeta)$ and $(\mathcal{R}_t, \mathcal{S}_t)$ are the ideal attribution functions induced by $(\mathsf{R}, \mathsf{S})$. By designing these pairs to be semantically tight, we reduce ambiguity and sharpen the interpretation of $\mathsf{Ver}(\zeta)$. In this relaxed setting, false negatives and false positives correspond to violations of the left and right inequalities, respectively.

In publicly verifiable settings, adversaries may have white-box access to $\mathsf{Ver}(\mathsf{pk}, \cdot)$ and attempt to induce false attribution. This motivates envelope guarantees that persist beyond the black-box regime, one for each side of the envelope: *unforgeability* (control over white-box false positives) and white-box *robustness* (control over white-box false negatives). Unforgeability requires that no efficient adversary, even given the detection key $\mathsf{pk}$, can produce a string that the verifier flags as attributable when it is not. This is the property one wants in practice to ensure that a positive flag $\mathsf{Ver}(\zeta) = 1$ constitutes credible evidence of provenance, since otherwise an adversary could falsely attribute fabricated text to the provider. While faithfulness already rules out forgeries by honest (black-box) users, unforgeability is the strictly stronger guarantee needed once the detection key is public. The ideal attribution framework makes this guarantee precise as control over the upper envelope $\mathsf{Ver}(\zeta) \leq \mathcal{S}_s(\zeta)$, a conceptually clean formulation made possible by specifying an explicit ideal target. We study unforgeable watermarking schemes in Section E.

## Acknowledgements

We thank Huijia (Rachel) Lin and Aloni Cohen for helpful discussions. This work was supported in part by the Simons Collaboration on the Theory of Algorithmic Fairness.

## Impact Statement

This work clarifies the semantics of false positives and false negatives in watermarking and AI-generated text detection by tying them to explicit ideal attribution targets. In current practice, error rates are often reported without precisely specifying what content is intended to be attributable, which can lead to ambiguous interpretations and, in some cases, false accusations of AI use against genuine authors whose writing styles happen to resemble those of deployed language models. By formalizing attribution mechanisms and watermarking schemes relative to explicit targets, our framework supports more careful assessment and responsible use of AI text and watermark detectors. In particular, attribution soundness (Definition 2.3) and faithfulness (Definition 3.1) guard against overconfident or misinterpreted ownership claims. We also note that a watermarking scheme's embedding capacity could in principle be repurposed by a model provider to carry unintended payloads, a covert channel risk that users of watermarked models should be aware of.

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

# A. Related work

**Watermarking schemes for language models.** Early work on text watermarking (Topkara et al., 2005; Atallah et al., 2001; 2002) focused on embedding signals into *fixed* objects in a way that is hard to remove without altering their perceived quality. Recent approaches treat language models as *randomized sampling algorithms* (Aaronson, 2022; Kirchenbauer et al., 2023a;b; Kuditipudi et al., 2023; Christ et al., 2024; Zhao et al., 2023), and embed watermarks by intervening in the sampling process, leveraging access to the original model's next-token probabilities. The intervention induces statistical correlations between the model's responses and a random watermark key fixed at initialization, enabling detection later from the response alone. For a broader overview of this emerging area, we refer the reader to the surveys (Liu et al., 2024; Zhao et al., 2025).

This work focuses on watermarking schemes that are undetectable and robust (Christ et al., 2024). Among existing undetectable schemes (Christ & Gunn, 2024; Cohen et al., 2024; Golowich & Moitra, 2024; Fairoze et al., 2025), the most relevant to our work is ideal pseudorandom codes (PRC) (Alrabiah et al., 2025). The notion of an *ideal attribution mechanism* is inspired by the ideal/real world paradigm of cryptographic security definitions, also made explicit in the formulation of ideal PRCs. In our view, pseudorandomness pertains to the distribution of the ledger rather than imperfections of the verifier in representing the target attribution mechanism. This perspective underlies our approach, which treats *undetectability* and *faithfulness* as formally distinct properties of watermarking schemes.

Another closely related work is that of Cohen et al. (2024), who introduce a conceptual framework for clarifying robustness guarantees in language model watermarking. Their AEB framework defines robustness through predicates on "blocks", where attribution depends on recovering enough blocks (or their approximations) from a candidate string. This yields a piecewise formulation in which robustness is expressed through distinct predicates, each defined for a particular transcript, rather than a single, well-defined attribution function determined by the ledger. Beyond this semantic difference, the AEB framework imposes a syntactic restriction requiring that, for any transcript, the blocks *partition* the model's response, disallowing overlaps. The ideal attribution framework relaxes this constraint by introducing attribution maps that do not enforce partitioning, thereby generalizing AEB syntactically.

**Pseudorandom codes.** Pseudorandom codes, introduced by Christ & Gunn (2024), are error-correcting codes whose codewords are pseudorandom to any observer lacking the decoding key. As objects that satisfy both pseudorandomness and robustness, PRCs provide a natural foundation for constructing watermarking schemes that achieve both undetectability and robustness (Christ & Gunn, 2024; Gunn et al., 2024; Cohen et al., 2024; Golowich & Moitra, 2024; Alrabiah et al., 2025). In fact, they can be viewed as watermarking schemes specialized to the uniform language model, but their versatility extends beyond this setting. Recently, PRCs were used to watermark generative *image* models (Gunn et al., 2024), yielding the first provably undetectable scheme for image models. This stands in contrast to other approaches to image watermarking, which target a more direct but less formal notion of watermark *invisibility*, evaluated empirically through extensive human indistinguishability experiments (see e.g., (Gowal et al., 2025)).

PRCs with varying robustness properties have been constructed under standard cryptographic assumptions, such as the subexponential hardness of learning parity with noise (LPN) (Christ & Gunn, 2024; Ghentiyala & Guruswami, 2025). In particular, Golowich & Moitra (2024) construct PRCs that are robust to a constant fraction of edits, including substitutions, insertions, and deletions, at the cost of requiring a larger alphabet.

However, the notion of robustness satisfied by previous PRC constructions is weaker than one might expect, as the decoder's guarantee holds only against corruptions generated *independently* of the PRC keys. In other words, the corruption adversary is assumed not to have observed any prior codewords. The absence of such *adaptive* robustness guarantees was highlighted by Cohen et al. (2024), who asked whether existing PRCs remain robust against adversaries that can observe previously generated codewords. Ideal PRCs, introduced by Alrabiah et al. (2025), address this limitation by constructing PRCs that remain robust even against adversaries with black-box access to both encoding and decoding oracles. This motivates our focus on ideal PRCs and the notion of faithfulness, which makes the subtle distinction between robustness notions explicit.

# B. Selection Rules

**Proposition B.1** (Surjection from selection rules to attribution maps)**.** *Every transcript-level attribution map* $\mathsf{R} : \{0,1\}^* \times \{0,1\}^* \to 2^{\{0,1\}^*}$ *satisfying Definition 2.1 can be induced by some selection rule* $\mathsf{Z} : \{0,1\}^* \times \{0,1\}^* \times \{0,1\}^* \to \{0,1\}$. *Conversely, any transcript-level attribution map satisfying Definition 2.1 induces a selection rule.*

*Proof.* Given a selection rule $\mathsf{Z}$, initialize with $\mathsf{R}(x, \square) := \emptyset$ and, for $u = u_1 \cdots u_\ell$, inductively define

$$\mathsf{R}(x, u_{1:j}) = \mathsf{R}(x, u_{1:j-1}) \cup \{u_{k:j} \mid 1 \le k \le j, \; \mathsf{Z}(x, u_{1:k-1}, \; u_{k:j}) = 1\} \,.$$

Conversely, given a transcript-level attribution map $\mathsf{R}$ and any $x, \rho, \zeta \in \{0,1\}^*$, define $\mathsf{Z}$ by

$$\mathsf{Z}(x, \rho, \zeta) \; = \; \mathbb{I}\left[\zeta \in \mathsf{R}(x, \rho\zeta) \setminus \mathsf{R}(x, \rho\zeta_{1:-1})\right] \,.$$

Starting from $\mathsf{R}$ satisfying Definition 2.1 and forming $\mathsf{Z}$ as above, the inductive construction recovers $\mathsf{R}$ since by the suffix-jump property of $\mathsf{R}$, for any $x \in \{0,1\}^*$ and $u \in \{0,1\}^*$,

$$
\begin{aligned}
\mathsf{R}(x, u_{1:j}) &= \mathsf{R}(x, u_{1:j-1}) \cup \Big(\mathsf{R}(x, u_{1:j}) \setminus \mathsf{R}(x, u_{1:j-1})\Big) \\
&= \mathsf{R}(x, u_{1:j-1}) \cup \{\zeta \in \mathsf{suffixes}(u_{1:j}) \mid \zeta \in \mathsf{R}(x, u_{1:j}) \setminus \mathsf{R}(x, u_{1:j-1})\} \\
&= \mathsf{R}(x, u_{1:j-1}) \cup \{u_{k:j} \mid 1 \le k \le j, \; \mathsf{Z}(x, u_{1:k-1}, \; u_{k:j}) = 1\} \,.
\end{aligned}
$$

Conversely, starting from an arbitrary selection rule $\mathsf{Z}$, it is straightforward to check that the induced $\mathsf{R}$ satisfies all the properties listed in Definition 2.1. $\qquad\square$

**Example B.2** (Non-injectivity). *Distinct selection rules may induce the same transcript-level attribution map. Fix any non-empty string $y \in \{0,1\}^*$. Define $\mathsf{Z}$ and $\mathsf{Z}'$ by*

$$\mathsf{Z}(x, \rho, \zeta) = \begin{cases} 1 & \text{if } (x, \rho, \zeta) \in \{(0, \square, y), (0, y, y)\} \,, \\ 0 & \text{otherwise} \,. \end{cases} \qquad \mathsf{Z}'(x, \rho, \zeta) = \begin{cases} 1 & \text{if } (x, \rho, \zeta) = (0, \square, y) \,, \\ 0 & \text{otherwise} \,. \end{cases}$$

*Although $\mathsf{Z}$ selects $y$ on input $(0, y, y)$, this extra selection does not cause changes in the attribution set since $y$ is already included after its first appearance. Thus, both rules induce the attribution set on any transcript $\pi$. Hence, $\mathsf{Z} \ne \mathsf{Z}'$ but $\mathsf{R}(\pi) = \mathsf{R}'(\pi)$ for all $\pi \in \{0,1\}^* \times \{0,1\}^*$.*

We now present sufficient (Lemma B.3) and necessary (Lemma B.5) conditions for a selection rule to satisfy attribution soundness. Given a language model $Q : \{0,1\}^* \to [0,1]$, a non-empty string $y \in \{0,1\}^*$, and a context $x \in \{0,1\}^*$, we define the *path-conditional measure* $Q(y \mid x)$ as the probability of autoregressively generating $y$ under the conditional distribution $Q(\cdot \mid x)$. Formally,

$$Q(y \mid x) = \prod_{j=1}^{\mathsf{len}(y)} Q(y_j \mid xy_{1:j-1}) \,.$$

Corollary B.4 shows that any selection rule which only selects text blocks with low path-conditional measure satisfies soundness. We note that the $\beta$-potential selection rule (Definition D.8) is stricter than the path-conditional measure selection rule. In other words, low predictive potential implies low path-conditional measure.

**Lemma B.3** (Path-conditional measure selection rule). *Let $\alpha > 0$ be a parameter and define the path-conditional measure selection rule by*

$$\mathsf{Z}(x, \rho, u \; ; Q) = \mathbb{I}\left[Q(u \mid x\rho) \le 2^{-\alpha}\right] \,.$$

*Then, for any $Q$, any fixed string $y \in \{0,1\}^*$, and any adversary observing length $T$ responses,*

$$\Pr\left[\mathcal{R}(y) = 1\right] \le T \cdot 2^{-\alpha} \,.$$

*Proof.* We prove the bound for a single transcript. The general case follows by the same argument. Let $y \in \{0,1\}^n$ be the fixed string. Define $E_t$ as the event string $y$ enters the attribution set at time $t$. That is,

$$
\begin{aligned}
E_t &= \mathbb{I}[y \in \mathsf{R}(\pi_t) \setminus \mathsf{R}(\pi_{t-1})] \\
&= (u_{t-n+1:t} = y) \wedge (\mathsf{Z}(x, u_{1:t-n}, y) = 1)
\end{aligned}
$$

Consider $t \geq n$. For any $\rho \in \{0,1\}^{t-n}$ such that $\mathsf{Z}(x, u_{1:t-n}, y) = 1$, we have $Q(E_t \mid x\rho) = Q(y \mid x\rho) \leq 2^{-\alpha}$. On the other hand, for any $\rho \in \{0,1\}^{t-n}$ such that $\mathsf{Z}(x, u_{1:t-n}, y) = 0$, we have $Q(E_t \mid x\rho) = 0$. Thus,

$$\Pr[E_t] = Q(E_t \mid x) = \sum_{\rho \in \{0,1\}^{t-n}} Q(E_t \mid x\rho) \cdot Q(\rho \mid x)$$
$$\leq 2^{-\alpha} \sum_{\rho \in \{0,1\}^{t-n}} Q(\rho \mid x)$$
$$\leq 2^{-\alpha} .$$

Applying the union bound over $t \in [T]$ gives

$$\Pr\left[\mathcal{R}(y) = 1\right] = \Pr\left[\bigvee_{t \leq T} E_t\right] \leq \sum_{t=1}^{T} \Pr[E_t] \leq T \cdot 2^{-\alpha} .$$

$\square$

**Corollary B.4** (Sufficient condition for attribution soundness). *Let $\lambda \in \mathbb{N}$ be the security parameter. For any language model $Q : \{0,1\}^* \to [0,1]$, any fixed sequence of strings $(y_\lambda)$, and any $T = \mathrm{poly}(\lambda)$, the attribution set $\mathsf{R}_\lambda(\Pi)$ induced by ledger $\Pi$ of length at most $T$, and the path-conditional measure selection rule $(\mathsf{Z}_\lambda)$ with parameter $\alpha = \omega(\log \lambda)$ satisfies*

$$\Pr\left[\Pi \leftarrow \mathcal{B}^{Q,\bar{Q}}(1^\lambda, y_\lambda) \; : \; y_\lambda \in \mathsf{R}_\lambda(\Pi)\right] \leq \mathrm{negl}(\lambda) .$$

Lemma B.5 establishes necessary conditions for selection rule soundness. It states that if a selection rule accepts (suffixes of) response strings with non-negligible path-conditional measure, then the corresponding attribution mechanism violates soundness.

**Lemma B.5** (Necessity of rejecting non-negligible blocks). *Let $\mathsf{Z} = (\mathsf{Z}_\lambda)$ be any sequence of selection rules, where each $\mathsf{Z}_\lambda$ is a deterministic function of the prompt $x \in \{0,1\}^*$, response prefix $\rho \in \{0,1\}^*$, response suffix $u \in \{0,1\}^*$, and the language model $Q : \{0,1\}^* \to [0,1]$.*

*Let $\eta \geq 1/\mathrm{poly}(\lambda)$ be a threshold parameter and $\ell = \mathrm{poly}(\lambda)$. Suppose there exist a language model $Q$, and sequences of prompts $(x_\lambda)$ and responses $(u_\lambda)$ of length at most $\ell$ such that*

$$Q(u_\lambda \mid x_\lambda) \geq \eta \quad \text{and} \quad \mathsf{Z}(x_\lambda, \rho_\lambda, y_\lambda; Q) = 1 ,$$

*where $u_\lambda = \rho_\lambda y_\lambda$, then the selection rule $\mathsf{Z}$ fails to satisfy attribution soundness (Definition 2.3).*

*Proof.* Let $Q$ be as above, and let $\bar{Q}(1^\lambda)$ denote the oracle that autoregressively samples length-$\ell(\lambda)$ responses from $Q$. Consider the adversary $\mathcal{B}$ that queries $\bar{Q}(1^\lambda)$ with the prompt $x_\lambda$ and generates a single transcript $\pi = (x_\lambda, u)$ where $u \leftarrow \bar{Q}(1^\lambda, x_\lambda)$. Then,

$$\Pr[y_\lambda \in \mathsf{R}_\lambda(\pi)] \geq \Pr[\rho_\lambda y_\lambda \sqsubseteq u]$$
$$= Q(\rho_\lambda y_\lambda \mid x_\lambda) \geq \eta ,$$

which is non-negligible in $\lambda$, violating attribution soundness. $\square$

## C. Necessary Conditions for Anytime Soundness

**Necessity of anytime soundness for faithfulness.** Let $Q$ be a language model, $\Phi$ a predicate, and $\mathsf{R} = (\mathsf{R}_\lambda)$ a sequence of transcript-level attribution maps. Suppose there exists a PPT adversary $\mathcal{A}$ and a polynomial $p : \mathbb{N} \to \mathbb{N}$ such that

$$\Pr\left[\zeta \leftarrow \mathcal{A}^{\bar{Q}}(1^\lambda) : \mathcal{R}_s^\Phi(\zeta) < \mathcal{R}_T^\Phi(\zeta)\right] > \frac{1}{p(\lambda)} ,$$

where $s$ is the time at which $\mathcal{A}$ outputs $\zeta$.

In other words, with noticeable probability, $\mathcal{A}$ produces a string $\zeta$ whose attribution decision with respect to $\mathsf{R}^{\Phi}$ differs between its generation time $s$ and the ledger termination time $T$.

Let $(\mathsf{Gen}, \mathsf{Wat}, \mathsf{Ver})$ be any undetectable and faithful watermarking scheme for $Q$, $\Phi$, and $\mathsf{R}$. By the undetectability of the watermarking scheme, we have that

$$\Pr \left[ \begin{array}{ll} (\mathsf{pk}, \mathsf{sk}) & \leftarrow \mathsf{Gen}(1^{\lambda}) \\ \zeta & \leftarrow \mathcal{A}^{\mathsf{Wat}_{\mathsf{sk}}}(1^{\lambda}) \end{array} : \mathcal{R}_s^{\Phi}(\zeta) < \mathcal{R}_T^{\Phi}(\zeta) \right] > \frac{1}{p(\lambda)} - \mathrm{negl}(\lambda) \,.$$

Now define a PPT adversary $\mathcal{A}'$ that simulates $\mathcal{A}$ and, upon obtaining $\zeta$, flips an independent fair coin. With probability $1/2$ it queries $\mathsf{Ver}_{\mathsf{pk}}(\zeta)$ at time $s$, the output time of $\mathcal{A}$, and with probability $1/2$ it queries at time $T$. By construction, with probability $1/2 \cdot (1/p(\lambda) - \mathrm{negl}(\lambda))$, $\mathcal{A}'$ satisfies

$$\mathsf{Ver}_{\mathsf{pk}}(\zeta) \neq \mathcal{R}_{\tau}^{\Phi}(\zeta) \,,$$

where $\tau \in \{s, T\}$ denotes the *random* query time chosen by $\mathcal{A}'$ according to the coin flip.

This contradicts the faithfulness guarantee of $\mathsf{Ver}_{\mathsf{pk}}$. Hence, anytime soundness is a necessary condition for faithfulness.

**Necessity of time coarsening for anytime soundness.** Anytime soundness is unachievable, even for the uniform language model, unless the global time set $\mathcal{T}$ is coarsened. Without coarsening, an adversary can efficiently recognize a response prefix on the edge of inclusion, i.e., a prefix that is one bit short of entering the attribution set, and exploit it to violate anytime soundness by issuing the attribution query as soon as the edge-of-inclusion event is observed.

Time coarsening prevents edge-of-inclusion attacks by restricting the set of allowed query times. The coarsened time set $\mathcal{T}' \subset \mathcal{T}$ ensures that adversaries cannot issue attribution queries exactly when edge-of-inclusion events occur (see Claim D.2).

**Example C.1** (Edge-of-inclusion adversary)**.** *Let $\ell = \mathrm{poly}(\lambda)$ be the response length and let $\mathsf{Z} = (\mathsf{Z}_{\lambda})$ be a non-vacuous sequence of selection rules such that for any constant $c > 0$, there exists $q = \mathrm{poly}(\lambda)$ such that*

$$\Pr \left[ \Pi_T \leftarrow \mathcal{U}_{\ell}^{\otimes q} : \mathsf{R}_{\lambda}(\Pi_T) \neq \emptyset \right] > 1 - c \,,$$

*where $\mathcal{U}_{\ell}$ denotes the uniform distribution over $\{0, 1\}^{\ell}$ and $T = (q, \ell)$.*

*We define the edge-of-inclusion adversary $\mathcal{A}$ as follows. For each transcript, it observes the growing response string $u_{1:j}$. For the first transcript index $i$ where it finds some $a \in \{0, 1\}$ such that*

$$\mathsf{R}(u_{1:j}) = \emptyset \quad and \quad \mathsf{R}(u_{1:j}a) \neq \emptyset \,,$$

*it outputs $\zeta = u_{1:j}a$ at time $s = (i, j)$ and continues the interaction until time $T$. We call $u_{1:j}$ the edge-of-inclusion string, and the time $s = (i, j)$ at which it is observed the edge-of-inclusion time.*

*Since the selection rule $\mathsf{Z}$ is non-vacuous, with probability at least $1 - c$ such an edge-of-inclusion event occurs before the termination time $T = \mathrm{poly}(\lambda)$. For $\zeta$ defined as such, we have*

$$\Pr \left[ \zeta \leftarrow \mathcal{A}^{\mathcal{U}_{\ell}}(1^{\lambda}) : \mathcal{R}_s(\zeta) = 0 \ \wedge \ \mathcal{R}_T(\zeta) = 1 \right] > \frac{1}{2}(1 - c) \,,$$

*which witnesses $\mathcal{R}_s(\zeta) \neq \mathcal{R}_T(\zeta)$ and thus violates anytime soundness.*

# D. Watermarking Schemes from Pseudorandom Codes

## D.1. Watermarking the uniform language model

We show that ideal PRCs yield undetectable and faithful watermarking schemes for the uniform language model $\mathcal{U}$, defined by $\mathcal{U}(x) = 1/2$ for all $x \in \{0, 1\}^*$. This simple, clean setting serves as a starting point for understanding what are achievable. We first identify the ideal attribution mechanism implicitly targeted by ideal PRCs, and then formally define ideal PRCs.

**Block-aligned attribution queries.** We consider faithfulness only at block-aligned attribution query times: Wat may generate responses token-by-token internally, but its outputs are revealed to the adversary in blocks of length $n$, and attribution queries occur only at these boundaries. All faithfulness statements henceforth are interpreted under this block alignment. See Section 3.1 for a discussion of why we restrict attention to coarse-grained attribution times.

**Block selection rule.** We define block selection rules $(\mathsf{Z}_\lambda)$ and coarse-grained time sets $(\mathcal{T}_\lambda)$, and show that they satisfy anytime soundness against adversaries time-aligned with $(\mathcal{T}_\lambda)$.

**Definition D.1** (Block selection and time blocks). *Fix a block length $n \in \mathbb{N}$. For any prompt $x \in \{0,1\}^*$, response prefix $\rho \in \{0,1\}^*$, and response suffix $\zeta \in \{0,1\}^*$, the* block selection rule $\mathsf{Z}_n$ *is defined by*

$$\mathsf{Z}_n(x, \rho, \zeta) = (\mathsf{len}(\rho) \equiv 0 \mod n) \wedge (\mathsf{len}(\zeta) = n) .$$

*The associated set of time blocks $\mathcal{T}_n \subset \mathcal{T}$ is*

$$\mathcal{T}_n = \{(i,j) \in \mathcal{T} \mid i \in \mathbb{N},\ j \equiv 0 \pmod{n}\} .$$

This follows the standard cryptographic convention of operating on *blocks* (i.e., fixed-length strings). Note that the block selection rule does not depend on the prompt $x$, which is natural since under the uniform language model the response distribution is invariant to the prompt.

When the block size scales as $n = \mathrm{poly}(\lambda)$, the corresponding Hamming-robust attribution mechanism satisfies anytime soundness (Claim D.2), a property necessary for the existence of faithful watermarking schemes. Thus, the ideal attribution mechanism defined by block selection rules and the Hamming predicate is a valid target for undetectable and faithful watermarking schemes. Ideal PRCs (Alrabiah et al., 2025) precisely yield undetectable and faithful watermarking schemes for this ideal attribution mechanism under the uniform language model.

**Claim D.2** (Anytime soundness under block selection). *Let $n = \mathrm{poly}(\lambda) \geq \lambda$, let $\mathcal{T}_\lambda \subset \mathcal{T}$ denote the coarsened time set with granularity $n$, consisting of time blocks of length $n$, let $\mathsf{Z}_\lambda$ be the block selection rule with block length $n$. For any fixed constant $\gamma < 1/2$, let $\Phi = (\mathsf{Ham}_{\gamma n})_{n \in \mathbb{N}}$.*

*Suppose $m = \mathrm{poly}(\lambda)$, and each transcript of the ledger-generating interaction has response length $\ell = nm$. Then, the corresponding ideal attribution mechanism with $(\mathsf{R}_\lambda^\Phi)$ for the uniform language model $\mathcal{U}$ satisfies anytime soundness with respect to $\mathcal{T}_n$.*

*Proof.* Condition on any transcript prefix up to a block boundary. Under the uniform language model, the next block is uniformly distributed over $\{0,1\}^n$ and is independent of past blocks. For any block $\zeta \in \{0,1\}^n$ that is not in $\mathcal{R}_s$, the event that $\zeta$ is $\Phi$-close to the next attribution set (i.e., is a soundness violation) at that time has probability at most $\mathrm{negl}(\lambda)$.

There are $m$ blocks per transcript and at most $\mathrm{poly}(\lambda)$ transcripts overall. A union bound over the at most $m \cdot r = \mathrm{poly}(\lambda)$ block steps shows that the probability of *any* violation across all transcripts and block times is still $\mathrm{negl}(\lambda)$. $\square$

**Ideal PRCs as watermarking schemes for $\mathcal{U}$.** Ideal PRCs are error-correcting codes whose codewords are computationally indistinguishable from the uniform distribution over $\{0,1\}^n$. Known constructions (Alrabiah et al., 2025) support robust decoding under perturbations in the Hamming metric (i.e., substitution errors).

**Definition D.3** (Ideal encoder and decoder). *Let $n = n(\lambda)$ denote the codeword size and $k = k(\lambda)$ denote the message size. The experiment maintains a global state, called the ledger $\Pi$, which logs all encoder outputs and their associated messages. The* ideal encoder and decoder *are oracles $(\mathcal{U}_n, \mathcal{R}^\Phi)$ defined as follows.*

- $\mathcal{U}_n(\sigma)$: *takes in a message $\sigma \in \{0,1\}^k$ and outputs a uniformly random block $y \in \{0,1\}^n$. The experiment appends the pair $(y, \sigma)$ to the ledger $\Pi$.*

- $\mathcal{R}^\Phi(\zeta)$: *takes in a block $\zeta \in \{0,1\}^n$ and outputs a message $\sigma \in \{0,1\}^k$ if there exists $(y, \sigma) \in \Pi$ such that $\zeta \in \Phi(y)$; otherwise, it returns $\perp$.*

**Definition D.4** (Ideal pseudorandom code). *For any predicate $\Phi$, an* ideal $\Phi$-robust pseudorandom code *with codeword size $n = n(\lambda)$, message size $k = k(\lambda)$ is a triple of PPT algorithms $(\mathsf{Gen}, \mathsf{Enc}, \mathsf{Dec})$ such that*

- $\mathsf{Gen}(1^\lambda)$: *generates random encoding and decoding keys* $(\mathsf{ek}, \mathsf{dk})$.

- $\mathsf{Enc}(\mathsf{ek}, \sigma)$: *takes in a message* $\sigma \in \{0,1\}^k$, *and outputs a random codeword* $\xi \in \{0,1\}^n$.

- $\mathsf{Dec}(\mathsf{dk}, \xi)$: *takes in a codeword* $\xi \in \{0,1\}^n$, *and outputs either a message* $\sigma \in \{0,1\}^k$ *or* $\perp$, *which denotes decoding failure*.

*and satisfy* ideal security*: for any PPT adversary* $\mathcal{A}$,

$$\left| \Pr[\mathcal{A}^{\mathcal{U}_n, \mathcal{R}^\Phi}(1^\lambda) = 1] - \Pr_{(\mathsf{ek}, \mathsf{dk}) \leftarrow \mathsf{Gen}(1^\lambda)}[\mathcal{A}^{\mathsf{Enc}_{\mathsf{ek}}, \mathsf{Dec}_{\mathsf{dk}}}(1^\lambda) = 1] \right| \le \mathrm{negl}(\lambda) \ ,$$

*where* $(\mathcal{U}_n, \mathcal{R}^\Phi)$ *is the* $\Phi$-*robust ideal encoder–decoder pair defined in Definition D.3*.

In words, ideal security means that no efficient adversary can distinguish whether it is interacting with the PRC encoder and decoder $(\mathsf{Enc}_{\mathsf{ek}}, \mathsf{Dec}_{\mathsf{dk}})$ or with their idealized counterparts $(\mathcal{U}_n, \mathcal{R}^\Phi)$. We refer to the special case $k = 0$ as a zero-bit PRC Christ & Gunn (2024, Section 1.2). In this case, the message space reduces to the singleton set $\{\square\}$ consisting of the empty string.

A zero-bit ideal PRC is precisely an undetectable and faithful watermarking scheme for the ideal attribution mechanism defined by the block selection rule. To make this correspondence explicit, let $\mathsf{PRC} = (\mathsf{Gen}, \mathsf{Enc}, \mathsf{Dec})$ be a zero-bit PRC. The corresponding watermarking scheme $(\mathsf{Gen}, \mathsf{Wat}, \mathsf{Ver})$ uses the same key generation algorithm $\mathsf{Gen}$, with watermarking keys defined as $\mathsf{sk} = \mathsf{ek}$ and $\mathsf{pk} = \mathsf{dk}$. The remaining components are defined as

$$\mathsf{Wat}_{\mathsf{sk}} = \mathsf{Enc}_{\mathsf{ek}}(\square) \quad \text{and} \quad \mathsf{Ver}_{\mathsf{pk}}(\zeta) = \mathbb{I}\left[\mathsf{Dec}_{\mathsf{dk}}(\zeta) \ne \perp\right] \ . \tag{1}$$

Each response block of $\mathsf{Wat}_{\mathsf{sk}}$ is a fresh PRC codeword. Ideal security of the PRC implies both the *undetectability* of $\mathsf{Wat}_{\mathsf{sk}}$ and its *faithfulness* to the target attribution mechanism. Specifically, ideal security guarantees that PRC codewords are pseudorandom, so the responses of $\mathsf{Wat}_{\mathsf{sk}}$ are computationally indistinguishable from responses from $\mathcal{U}_n$. Moreover, it implies faithfulness as shown in Lemma D.5. Note that the interacting PPT adversary $\mathcal{A}$ can, in principle, maintain the ledger of the interaction explicitly and thus simulate the ideal decoder $\mathcal{R}^\Phi$ on its own. Hence, oracle access to $\mathcal{R}^\Phi$ is actually redundant.

**Lemma D.5** (Ideal security implies faithfulness). *Assume ideal security, i.e., for any PPT* $\mathcal{A}$,

$$\left| \Pr[\mathcal{A}^{\mathsf{Wat}_{\mathsf{sk}}, \mathsf{Ver}_{\mathsf{pk}}}(1^\lambda) = 1] - \Pr[\mathcal{A}^{\mathcal{U}_n, \mathcal{R}^\Phi}(1^\lambda) = 1] \right| \le \mathrm{negl}(\lambda) \ .$$

*Then for any PPT* $\mathcal{A}$,

$$\left| \Pr[\mathcal{A}^{\mathsf{Wat}_{\mathsf{sk}}, \mathcal{R}^\Phi}(1^\lambda) = 1] - \Pr[\mathcal{A}^{\mathsf{Wat}_{\mathsf{sk}}, \mathsf{Ver}_{\mathsf{pk}}}(1^\lambda) = 1] \right| \le \mathrm{negl}(\lambda) \ .$$

*Proof.* Fix any PPT adversary $\mathcal{A}$ and define the Bernoulli variables

$$H_0 = \mathcal{A}^{\mathsf{Wat}_{\mathsf{sk}}, \mathsf{Ver}_{\mathsf{pk}}}(1^\lambda) \ , \qquad\qquad H_1 = \mathcal{A}^{\mathcal{U}_n, \mathcal{R}^\Phi}(1^\lambda) \ , \qquad\qquad H_2 = \mathcal{A}^{\mathsf{Wat}_{\mathsf{sk}}, \mathcal{R}^\Phi}(1^\lambda) \ .$$

By ideal security of the PRC,

$$\left| \Pr[H_0 = 1] - \Pr[H_1 = 1] \right| \le \mathrm{negl}(\lambda) \ .$$

The ledgers generated by $\mathsf{Wat}_{\mathsf{sk}}$ and $\mathcal{U}_n$ are computationally indistinguishable. Moreover, a PPT adversary can simulate each ideal attribution function directly from the ledger it maintains during interaction, so oracle access to $\mathcal{R}^\Phi$ offers no additional power. Therefore,

$$\left| \Pr[H_1 = 1] - \Pr[H_2 = 1] \right| \le \mathrm{negl}(\lambda) \ .$$

By the triangle inequality,

$$\left| \Pr[H_0 = 1] - \Pr[H_2 = 1] \right| \le \left| \Pr[H_0 = 1] - \Pr[H_1 = 1] \right| + \left| \Pr[H_1 = 1] - \Pr[H_2 = 1] \right| \le \mathrm{negl}(\lambda) \ .$$

$\square$

From Lemma D.5, we obtain the following corollary.

**Corollary D.6.** *Let $\Phi$ be any efficiently computable predicate. The watermarking scheme constructed from a zero-bit $\Phi$-robust ideal PRC, as defined in Eq. (1), is undetectable under the uniform language model $\mathcal{U}$, and faithful with respect to the $\Phi$-robust attribution mechanism induced by the block selection rule (Definition D.1).*

**Remark D.7** (Variable-length string). *For candidate strings $\zeta \in \{0,1\}^*$ longer than the minimal block size $n$, attribution can be performed by applying a length-$n$ sliding window over $\zeta$ and checking each block individually.*

### D.2. Watermarking general language models

We consider PRC-based watermarking schemes for general language models proposed by Christ & Gunn (2024). In Theorem D.10, we show that the scheme satisfies undetectability and robustness guarantees with respect to an ideal attribution mechanism defined by the selection rule based on predictive potential (Definition D.8). The scheme, however, fails to satisfy soundness against black-box adversaries and therefore falls short of faithfulness (Example D.12). Moreover, this gap is fundamental for autoregressive schemes with irrevocable commitment: no such scheme that is faithful under the uniform model can remain faithful under all general language models (Lemma D.14).

**Definition D.8** (Predictive potential). *Let $Q : \{0,1\}^* \to [0,1]$ be a language model, $\rho \in \{0,1\}^*$ a context, and $y \in \{0,1\}^n$ a response block. The* predictive potential *of $y$ with respect to model $Q$ and context $\rho$ is defined as*

$$\mathsf{B}_n(y; \rho, Q) = \sum_{j=1}^{n} \left| \frac{1}{2} - Q(y_j \mid \rho y_{<j}) \right| .$$

**Definition D.9** ($\beta$-potential block selection). *Let $\beta \in (0, 1/2)$ be a fixed constant, and let $n \in \mathbb{N}$ denote the block length. The $\beta$-potential selection rule is defined as the conjunction of the block selection rule $\mathsf{Z}_n^{\mathsf{block}}$ (Definition D.1) and a bounded predictive potential condition:*

$$\mathsf{Z}_n(x, \rho, \zeta; Q) = \mathsf{Z}_n^{\mathsf{block}}(x, \rho, \zeta) \wedge \left( \mathsf{B}_n(\zeta; x\rho, Q) \leq \beta n \right) .$$

When $Q = \mathcal{U}$, the potential-based selection rule reduces to the block selection rule $\mathsf{Z}_n^{\mathsf{block}}$.

**Theorem D.10** (PRC-based watermarking under ideal PRCs). *Let $Q$ be any language model, and let $\Phi$ be an efficiently computable predicate. Let $\gamma \in [0, 1/4)$ be a constant, and let $\mathsf{PRC}[\mathsf{Gen}, \mathsf{Enc}, \mathsf{Dec}]$ be a zero-bit ideal PRC that is robust with respect to the composite predicate $\Phi \circ \mathsf{Ham}_{\gamma n}$.*

*Then, for any constant $\beta \geq 0$ satisfying $\beta < \gamma$, or $\beta = \gamma = 0$, the $\mathsf{PRC}$-based watermarking scheme $(\mathsf{Gen}, \mathsf{Wat}, \mathsf{Ver})$ (Algorithm 1) is* undetectable *and* robust *with respect to the ideal $\Phi$-robust attribution mechanism defined by the $\beta$-potential selection rule.*

We now describe the construction of the PRC-based watermarking scheme. The only modification from the uniform language model case in Section D.1 lies in the watermarking algorithm $\mathsf{Wat}$; the key generation $\mathsf{Gen}$ and verification $\mathsf{Ver}$ remain unchanged (see Eq. (1)). The central component of $\mathsf{Wat}$ is a *randomized* embedding (Christ & Gunn, 2024, Algorithm 2)

$$\mathsf{Embed} : \{0,1\} \times [0,1] \to \{0,1\} , \tag{2}$$

which maps a source bit to a watermarked bit.

**Embedding guarantees and undetectability.** The embedding satisfies the following properties. Let $Q : \{0,1\}^* \to [0,1]$ be the language model and let $\rho \in \{0,1\}^*$ be the preceding context. If $U \leftarrow \mathrm{Ber}(1/2)$ and $Y \leftarrow \mathsf{Embed}(U, Q(\rho))$, then

$$\Pr[Y = 1] = Q(\rho) ,$$
$$\Pr[Y = U] = 1 - |Q(\rho) - 1/2| .$$

Observe that the pushforward of the uniform distribution over $\{0,1\}^n$ under the autoregressive application of $\mathsf{Embed}$ starting from context $\rho$ satisfies

$$\mathsf{Embed}_\sharp \mathcal{U}_n = \bar{Q}_n(\rho) ,$$

where $\mathcal{U}_n$ is the uniform distribution over $\{0,1\}^n$, and $\bar{Q}_n(\rho)$ is the distribution induced by autoregressive sampling from $Q$ for $n$ steps given context $\rho$. When the source bits are pseudorandom, the pushforward condition guarantees computational indistinguishability from $\bar{Q}_n(\rho)$, and hence watermark undetectability.

---

**Algorithm 1** Watermarked response generation via $\mathsf{Wat}_{\mathsf{sk}}(\cdot)$

---

1: **input:** language model $Q$, security parameter $\lambda$, PRC encoding key ek, prompt $x$
2: $n \leftarrow n(\lambda), m \leftarrow m(\lambda)$             `// block length and number of blocks per response`
3: $\xi \leftarrow \mathsf{PRC.Enc}_{\mathsf{ek}}(\square)$                  `// sample initial PRC codeword of length n`
4: $i, j, k \leftarrow 1$
5: **while** $k \leq m$ **do**
6:     `// tokenwise embedding step`
7:     $p_i \leftarrow Q(xa_{1:i-1})$
8:     $a_i \leftarrow \mathsf{Ber}(p_i - (-1)^{\xi_j} \cdot \min\{p_i, 1 - p_i\})$        `// embed PRC bit into model response`
9:     $i \leftarrow i + 1, j \leftarrow j + 1$
10:    **if** $j > n$ **then**
11:       `// end of current PRC block`
12:       $\xi \leftarrow \mathsf{PRC.Enc}_{\mathsf{ek}}(\square)$
13:       $j \leftarrow 1, k \leftarrow k + 1$
14:    **end if**
15: **end while**
16: **return** $a_{1:mn}$

---

**Generation process of** $\mathsf{Wat}_{\mathsf{sk}}$ **and embedding noise.** $\mathsf{Wat}_{\mathsf{sk}}$ first samples a PRC codeword $\xi \leftarrow \mathsf{Enc}(\square)$ of length $n$, which provides the pseudorandom source bits for the embedding procedure Embed. The model then generates tokens autoregressively using Embed, consuming one PRC bit at a time while updating the reference context $\rho$. When the source PRC bits are exhausted, a new codeword is sampled to provide for the next block. A pseudocode description is provided in Algorithm 1.

The embedding introduces substitution noise relative to the source codeword $\xi$. The following lemma, a simple consequence of a multiplicative Chernoff bound (see, e.g., (Vershynin, 2018, Theorem 2.3.1)), shows that whenever the total expected noise is at most $\beta n$ (as ensured by the $\beta$-potential selection rule), the realized Hamming error remains below $\gamma n$ with overwhelming probability.

**Lemma D.11** (Hamming error of embedding channel). *Let $\beta < \gamma$ be fixed constants. Let $Z_1, \ldots, Z_n$ be independent Bernoulli variables such that $\mathbb{E}[Z_i] = p_i$ with $\sum_{i=1}^{n} p_i \leq \beta n$. Then there exists a constant $c = c(\beta, \gamma) > 0$ such that*

$$\Pr\left[\sum_{i=1}^{n} Z_i \geq \gamma n\right] \leq e^{-cn} .$$

Combining the pushforward property of the embedding (for undetectability) with the bounded noise guarantee of Lemma D.11 and the robustness of the ideal PRC immediately yields Theorem D.10.

**Weakened faithfulness under general models.** If $Q(\rho) = 1/2$, then $Y = U$ with probability 1. Hence, if $Q = \mathcal{U}$, the resulting watermarking scheme exactly recovers the scheme of Section D.1. For general language models this equality no longer holds; Embed introduces substitution noise in generating the response, so the target attribution mechanism must be more conservative.

In Theorem D.10, the target attribution is robust only with respect to $\Phi$, whereas the underlying PRC is robust with respect to the composite predicate $\Phi \circ \mathsf{Ham}_{\gamma n}$. Thus robustness is preserved, but the guarantee against false positives is weakened. The next example shows how a black-box adversary can exploit this.

**Example D.12** (False positives under a conservative target). *Let $\delta > \gamma > 0$ be fixed constants, and assume that the underlying ideal PRC satisfies ideal security with respect to the predicate $\mathsf{Ham}_{\delta n}$. Define $\Phi = \mathsf{Ham}_{(\delta-\gamma)n}$ so that $\mathsf{Ham}_{\delta n} = \Phi \circ \mathsf{Ham}_{\gamma n}$, as in Theorem D.10.*

*Consider a prompt $x \in \{0,1\}^*$ such that the conditional response distribution $Q(\cdot \mid x)$ is uniform. For example, $x =$ "generate uniformly random bits: ". For such prompts, $\mathsf{Wat}$ outputs an uncorrupted PRC codeword. An adversary can sample a response $y \leftarrow \mathsf{Wat}_{\mathsf{sk}}(x)$, and apply a perturbation of magnitude $(\delta - \gamma/2)n$ to obtain a perturbed response $y'$. By construction, $y'$ remains within $\mathsf{Ham}_{\delta n}$ of the original PRC codeword and therefore satisfies $\mathsf{Ver}_{\mathsf{pk}}(y') = 1$. However, $y'$*

*lies outside the $\Phi$-expansion of $y$, and hence $\mathcal{R}_t^\Phi(y') = 0$ for the target attribution mechanism induced by the $\beta$-potential rule (for any $\beta \in [0,1]$).*

**Definition D.13** (Autoregressive sampler with irrevocable commitment). *A watermarking sampler* Wat *is autoregressive with irrevocable commitment if, on input a prompt $x$ and key* wk*, it generates response $y = (y_1, \ldots, y_n)$ sequentially: at each step $j$, it queries the conditional $Q(\cdot \mid x, y_{<j})$, outputs $y_j$, and irrevocably commits to $y_j$ before proceeding to the next step. In particular, the distribution of the emitted prefix $y_{<j}$ depends on the reference model $Q$ only through the conditionals $\{Q(\cdot \mid x, y_{<i})\}_{i<j}$.*

**Lemma D.14** (Faithfulness gap for irrevocable samplers). *There exist a selection rule* Z *and a language model $Q$ such that for any watermarking scheme* (Gen, Wat, Ver) *with* Wat *undetectable, autoregressive, and irrevocable (Definition D.13), if the scheme is faithful under the uniform language model $U$, then it is not faithful under $Q$.*

*Proof.* Let $n = n(\lambda) \geq \lambda$, and let $\{f_{\mathsf{sk}} \colon \{0,1\}^{n-1} \to \{0,1\}\}_{\mathsf{sk}}$ be a pseudorandom function (PRF) family. The seed sk is sampled independently and kept hidden from Ver. Let $Q_{\mathsf{sk}}$ be a language model defined by

$$Q_{\mathsf{sk}}(u_j \mid x, u_{<j}) = \begin{cases} 1/2 & \text{if } j < n, \\ \mathbf{1}[u_j = f_{\mathsf{sk}}(u_1, \ldots, u_{n-1})] & \text{if } j = n. \end{cases}$$

Let Z be the 0-potential block selection rule,

$$\mathsf{Z}(x, \rho, \zeta) = (\mathsf{len}(\rho) \equiv 0 \pmod{n}) \wedge (\mathsf{len}(\zeta) = n) \wedge (B_n(\zeta; x\rho, Q) = 0).$$

**Ideal attribution under each model.** Under the uniform language model $U$, every bit of a block is sampled with conditional probability $1/2$, so $B_n = 0$ and Z selects every emitted block; faithfulness then requires $\mathsf{Ver}_{\mathsf{wk}}(Y) = 1$ with overwhelming probability over $Y \leftarrow \mathsf{Wat}_{\mathsf{wk}}^U$. Under $Q_{\mathsf{sk}}$, the $n$-th bit is deterministic, so $B_n > 0$ and Z rejects all blocks, giving an empty attribution set; faithfulness then requires $\mathsf{Ver}_{\mathsf{wk}}(Y') = 0$ with overwhelming probability over $Y' \leftarrow \mathsf{Wat}_{\mathsf{wk}}^{Q_{\mathsf{sk}}}$.

**Prefix distribution under both models.** At positions $j < n$, the conditional $Q_{\mathsf{sk}}(\cdot \mid x, y_{<j}) = 1/2$ is identical under $U$ and $Q_{\mathsf{sk}}$. By irrevocable commitment, the emitted prefix $(Y_1, \ldots, Y_{n-1})$ depends on the reference model only through these positions, so its distribution under $\mathsf{Wat}_{\mathsf{wk}}$ is the same for both $U$ and $Q_{\mathsf{sk}}$. Since $Q_{\mathsf{sk}}$'s position-$n$ conditional is deterministic, undetectability of the scheme under $Q_{\mathsf{sk}}$ implies that the emitted final bit agrees with it, $Y_n = f_{\mathsf{sk}}(Y_1, \ldots, Y_{n-1})$, with overwhelming probability over wk and the sampling randomness of $\mathsf{Wat}_{\mathsf{wk}}^{Q_{\mathsf{sk}}}$.

**Contradiction via PRF security.** Fix wk. For each prefix $\rho \in \{0,1\}^{n-1}$ define the acceptance set

$$A_{\mathsf{wk}}(\rho) = \{ a \in \{0,1\} : \mathsf{Ver}_{\mathsf{wk}}(\rho a) = 1 \},$$

which is efficiently computable from wk alone and depends only on $\rho$ and wk, not on which language model was used as reference to generate $\rho$.

Suppose the scheme is faithful under both $U$ and $Q_{\mathsf{sk}}$. Consider a single run of $\mathsf{Wat}_{\mathsf{wk}}^{Q_{\mathsf{sk}}}$, producing an emitted block with prefix $\rho = Y_{1:n-1}$. With overwhelming probability over this run:

- By the prefix argument, $Y_n = f_{\mathsf{sk}}(\rho)$, so the emitted block is $\rho f_{\mathsf{sk}}(\rho)$. Faithfulness under $Q_{\mathsf{sk}}$ applies to this emitted block; since Z rejects it, faithfulness forces $\mathsf{Ver}_{\mathsf{wk}}(\rho f_{\mathsf{sk}}(\rho)) = 0$, i.e. $f_{\mathsf{sk}}(\rho) \notin A_{\mathsf{wk}}(\rho)$.

- Since the prefix distributions of $\mathsf{Wat}_{\mathsf{wk}}^U$ and $\mathsf{Wat}_{\mathsf{wk}}^{Q_{\mathsf{sk}}}$ coincide, faithfulness under $U$ implies that, with overwhelming probability over this same prefix $\rho$, at least one continuation is accepted; that is, $A_{\mathsf{wk}}(\rho) \neq \emptyset$.

Combining the two, $A_{\mathsf{wk}}(\rho)$ is nonempty yet excludes $f_{\mathsf{sk}}(\rho)$, so it is the singleton $\{1 - f_{\mathsf{sk}}(\rho)\}$.

Crucially, the watermark key wk is generated by $\mathsf{Gen}(1^\lambda)$, independently of the PRF seed sk; the seed enters only through the reference model $Q_{\mathsf{sk}}$. Thus the map $\rho \mapsto 1 - a$, where $A_{\mathsf{wk}}(\rho) = \{a\}$, is computable from wk alone, without sk or even oracle access to $f_{\mathsf{sk}}$.

This contradicts the unpredictability of $f$. Indeed, the shared prefix distribution $\mu_{\mathsf{wk}}$ is efficiently sampleable from wk and independent of sk, so no efficient predictor computed from wk alone can predict $f_{\mathsf{sk}}(\rho)$ for $\rho \leftarrow \mu_{\mathsf{wk}}$ with non-negligible advantage over $1/2$. But the function $\rho \mapsto 1 - a$, where $A_{\mathsf{wk}}(\rho) = \{a\}$, predicts $f_{\mathsf{sk}}(\rho)$ with overwhelming probability. Hence the scheme cannot be faithful under $Q_{\mathsf{sk}}$. $\qquad\square$

# E. Beyond Black-Box Adversaries

We study faithfulness guarantees that serve as goals for future watermarking schemes. In particular, we focus on guarantees against *white-box* adversaries, i.e., adversaries with direct access to the detection key $\mathsf{pk}$ and therefore white-box access to the detector $\mathsf{Ver}(\mathsf{pk}, \cdot)$. Such guarantees are essential for publicly verifiable watermarking schemes. In Sections E.1 and E.2, we focus on the unforgeability guarantees of existing schemes (Christ & Gunn, 2024; Fairoze et al., 2025), which build on digital signatures. We emphasize that these guarantees are limited, as they lack robustness to perturbations of the signed response. These observations motivate the search for a notion of *robust* digital signature that remains verifiable under mild perturbations while retaining a suitably relaxed form of unforgeability. We leave this investigation to future work.

**Unforgeability and robustness.** The failure event in relaxed faithfulness can be expressed as the union of two disjoint cases: false positives and false negatives,

$$\neg\big(\mathcal{R}_s(\zeta) \leq \mathsf{Ver}_{\mathsf{pk}}(\zeta) \leq \mathcal{S}_s(\zeta)\big) = \big(\mathsf{Ver}_{\mathsf{pk}}(\zeta) > \mathcal{S}_s(\zeta)\big) \vee \big(\mathsf{Ver}_{\mathsf{pk}}(\zeta) < \mathcal{R}_s(\zeta)\big) .$$

A false positive occurs when $\mathsf{Ver}_{\mathsf{pk}}$ accepts a string $\zeta$ not actually attributable under $\mathcal{R}_s$, for instance a string that is far from all responses generated by $\mathsf{Wat}_{\mathsf{sk}}$. If an adversary can reliably construct such strings, the scheme is vulnerable to *forgery*. Forgeries undermine the credibility of watermarks, as they allow non-attributable text to be falsely flagged as watermarked. While faithfulness ensures that honest users interacting with $\mathsf{Ver}_{\mathsf{pk}}$ in a black-box manner cannot produce such forgeries, malicious adversaries may attempt to exploit the public key $\mathsf{pk}$.

*Unforgeability* guarantees that false positives cannot be produced even by adversaries with direct access to $\mathsf{pk}$. This ensures that the verifier cannot be fooled into accepting non-attributable strings, thereby strengthening the credibility of an attribution decision $\mathsf{Ver}_{\mathsf{pk}}(\zeta) = 1$.

A false negative, by contrast, occurs when the verifier rejects a string that is genuinely attributable under $\mathcal{R}_s$. Here we abuse notation slightly and write $\mathcal{R}_s$ for the robust attribution function (previously denoted $\mathcal{R}_s^{\Phi}$, with the predicate $\Phi$ made explicit). *Robustness* provides protection against such failures. While robustness can be ensured in the black-box setting, whether it can be achieved against white-box adversaries remains open. Existing constructions, even when specialized to the uniform language model, are provably non-robust against white-box adversaries (Alrabiah et al., 2025, Footnote 7). Addressing this limitation remains an interesting direction for future work.

## E.1. Unforgeable schemes for the uniform language model

We describe an unforgeable watermarking scheme for the uniform language model $\mathcal{U}$ obtained by composing a digital signature scheme with a multi-bit ideal PRC. The construction is a slight variant of the scheme introduced by Christ & Gunn (2024). Further background on digital signature schemes is provided in Section F.

**Construction of an unforgeable scheme.** Let $\mathsf{DS}[\mathsf{Gen}, \mathsf{Sign}, \mathsf{Ver}]$ be a digital signature scheme with signature size $k$, and let $\mathsf{PRC}[\mathsf{Gen}, \mathsf{Enc}, \mathsf{Dec}]$ be an ideal PRC with respect to predicate $\Phi$, with codeword size $n$ and message size $k$. We define the watermarking scheme $(\mathsf{Gen}, \mathsf{Wat}, \mathsf{Ver})$ as follows. First, generate the component keys

$$(\mathsf{DS.pk}, \mathsf{DS.sk}) \leftarrow \mathsf{DS.Gen}(1^\lambda) ,$$
$$(\mathsf{PRC.ek}, \mathsf{PRC.dk}) \leftarrow \mathsf{PRC.Gen}(1^\lambda) .$$

The watermark keys are then defined as

$$\mathsf{pk} = (\mathsf{DS.pk}, \mathsf{PRC.ek}, \mathsf{PRC.dk}) ,$$
$$\mathsf{sk} = \mathsf{DS.sk} .$$

Within a single transcript, $\mathsf{Wat}_{\mathsf{sk}}$ first samples a random message $\sigma_0 \leftarrow \{0,1\}^k$ and encodes it into the initial PRC block

$$\xi_1 \leftarrow \mathsf{PRC.Enc}(\mathsf{PRC.ek}, \sigma_0) .$$

Subsequent response blocks are then generated sequentially, embedding the signature of each response block into the next:

$$\sigma_i \leftarrow \mathsf{DS.Sign}(\mathsf{DS.sk}, \xi_i) ,$$
$$\xi_{i+1} \leftarrow \mathsf{PRC.Enc}(\mathsf{PRC.ek}, \sigma_i) .$$

Given a string $\zeta \in \{0,1\}^{2n}$, written as $\zeta = \zeta_1\zeta_2$ with each component block of length $n$, verification proceeds by PRC-decoding the suffix and checking whether the recovered signature validates the prefix.

$$\mathsf{Ver}(\mathsf{pk}, \zeta_1\zeta_2) = \mathsf{DS.Ver}(\mathsf{DS.pk}, \zeta_1, \sigma) \ ,$$
$$\text{where } \sigma = \mathsf{PRC.Dec}(\mathsf{PRC.dk}, \zeta_2) \ .$$

**Target ideal mechanisms.** Let $\mathsf{Z}_n^{\mathsf{DS}}$ denote the following modified block selection rule

$$\mathsf{Z}_n^{\mathsf{DS}}(x, \rho, \zeta) = (\mathsf{len}(\rho) \equiv 0 \mod n) \wedge (\mathsf{len}(\zeta) = 2n) \ . \tag{3}$$

Let $(\mathcal{R}_t)$ denote the corresponding sequence of (non-robust) ideal attribution functions. The scheme is unforgeable with respect to the following induced attribution function:

$$\mathcal{S}_t(\zeta_1\zeta_2) = 1 \iff \exists u \in \{0,1\}^n \text{ such that } \mathcal{R}_t(\zeta_1 u) = 1 \ . \tag{4}$$

Intuitively, $\mathcal{S}_t(\zeta) = 1$ if the first half of $\zeta \in \{0,1\}^{2n}$ appears as the prefix of some length-$2n$ response block within the transcript $\Pi_t$.

**Robustness predicate.** Due to the fragility of the digital signature, the scheme achieves only a weakened form of robustness compared to the underlying PRC, captured by the following modified predicate $\varphi$. For any strings $y, y' \in \{0,1\}^{2n}$, written as $y = y_1y_2$ and $y' = y_1'y_2'$ with $y_1, y_2, y_1', y_2' \in \{0,1\}^n$,

$$\varphi(y, y') = (y_1 = y_1') \wedge \Phi(y_2, y_2') \ . \tag{5}$$

Equivalently, in terms of predicate expansion,

$$\varphi(y_1y_2) = \{y_1 u \mid u \in \Phi(y_2)\} \ .$$

Note that for any $\zeta \in \{0,1\}^{2n}$ and $t \in \mathcal{T}$,

$$\mathcal{R}_t^{\varphi}(\zeta) \leq \mathcal{S}_t(\zeta) \ .$$

Hence, $(\mathcal{R}_t^{\varphi}, \mathcal{S}_t)$ forms a valid envelope for potentially sandwiching $\mathsf{Ver}_{\mathsf{pk}}$.

The following theorem formalizes the guarantees of the watermarking scheme. Its proof follows directly from the strong unforgeability of the underlying digital signature scheme and the ideal security of the PRC.

**Theorem E.1** (Unforgeable watermarking, uniform model). *Let* $\mathsf{PRC}[\mathsf{Gen}, \mathsf{Enc}, \mathsf{Dec}]$ *be an ideal PRC with respect to predicate* $\Phi$, *with codeword size* $n$ *and message size* $k$, *and let* $\mathsf{DS}[\mathsf{Gen}, \mathsf{Sign}, \mathsf{Ver}]$ *be a digital signature scheme with signature size* $k$.

*The watermarking scheme* $(\mathsf{Gen}, \mathsf{Wat}, \mathsf{Ver})$ *described above is undetectable with respect to the uniform language model* $\mathcal{U}$ *and satisfies the following faithfulness guarantees.*

**Unforgeability:** *For any white-box PPT adversary* $\mathcal{A}$ *that outputs a string in* $\{0,1\}^{2n}$,

$$\Pr\left[ \begin{array}{ll} (\mathsf{pk}, \mathsf{sk}) & \leftarrow \mathsf{Gen}(1^\lambda) \\ \zeta & \leftarrow \mathcal{A}^{\mathsf{Wat}_{\mathsf{sk}}, \mathsf{Ver}_{\mathsf{pk}}}(1^\lambda, \mathsf{pk}) \end{array} : \mathsf{Ver}(\mathsf{pk}, \zeta) > \mathcal{S}_s(\zeta) \right] \leq \mathrm{negl}(\lambda) \ ,$$

*where* $s \in \mathcal{T}$ *denotes the time at which* $\zeta$ *is output (the adversary may continue interacting after this point), and* $\mathcal{S}_s$ *is the ideal attribution function defined in Eq.* (4).

**Faithfulness:** *For any black-box PPT adversary* $\mathcal{A}$ *that outputs a string in* $\{0,1\}^{2n}$,

$$\Pr\left[ \begin{array}{ll} (\mathsf{pk}, \mathsf{sk}) & \leftarrow \mathsf{Gen}(1^\lambda) \\ \zeta & \leftarrow \mathcal{A}^{\mathsf{Wat}_{\mathsf{sk}}, \mathsf{Ver}_{\mathsf{pk}}}(1^\lambda) \end{array} : \mathsf{Ver}(\mathsf{pk}, \zeta) \neq \mathcal{R}_s^{\varphi}(\zeta) \right] \leq \mathrm{negl}(\lambda) \ ,$$

*where* $s \in \mathcal{T}$ *denotes the time at which* $\zeta$ *is output (the adversary may continue interacting after this point),* $\varphi$ *denotes the modified predicate defined in Eq.* (5)*, and* $\mathcal{R}_s$ *denotes the ideal attribution function induced by the modified block selection rule defined in Eq.* (3).

**Motivation for *robust* digital signatures.** Robustness of the above unforgeable scheme, represented by the predicate $\varphi$ (see Eq. (5)), is unsatisfactory because $\varphi$ does not tolerate *any* perturbations to the prefix block. This is a direct consequence of the inherent fragility of digital signatures. This limitation, also present in the unforgeable scheme of Fairoze et al. (2025), motivates the development of a *robust* digital signature that remains verifiable under mild perturbations while supporting a modified notion of unforgeability. We conjecture that such primitives can be constructed using ideas from secure sketches (Dodis et al., 2008).

### E.2. Unforgeable schemes for general language models

For general language models $Q$, we define the target attribution mechanism using a modified $\beta$-potential selection rule (Definition D.8). For any string $\zeta \in \{0,1\}^*$ that can be split into equal-length prefix and suffix blocks $\zeta_1$ and $\zeta_2$, define

$$\mathsf{Z}_n(x,\rho,\zeta_1\zeta_2) = \mathsf{Z}_n^{\mathsf{DS}}(x,\rho,\zeta_1\zeta_2) \,\wedge\, (\mathsf{B}_n(\zeta_1; x\rho, Q) \leq \beta n) \,\wedge\, (\mathsf{B}_n(\zeta_2; x\rho\zeta_1, Q) \leq \beta n) \,. \tag{6}$$

This rule restricts attribution to blocks whose prefix and suffix are sufficiently *unpredictable* under $Q$. The watermarking scheme targeting this attribution mechanism follows the same structure as the uniform case described in Section E.1, with one key modification: instead of signing the codeword $\xi_i$, the watermarking algorithm $\mathsf{Wat}_{\mathsf{sk}}$ signs the generated response block $y_i$ produced by the embedding map Embed Eq. (2), and does so only if its predictive potential is low.

More precisely,

$$\sigma_i \leftarrow \begin{cases} \mathsf{DS.Sign}(\mathsf{DS.sk}, y_i) & \text{if } \mathsf{B}_n(y_i; x\rho, Q) \leq \beta n \,, \\ \{0,1\}^k & \text{otherwise} \,. \end{cases}$$

When $Q = \mathcal{U}$, the watermarking scheme and its target attribution mechanism reduce exactly to the uniform case analyzed in Section E.1. The following theorem formalizes the guarantees of the scheme. As before, no robustness guarantees are provided against perturbations to the prefix block, since even the upper envelope $\mathcal{S}_t$ for $\mathsf{Ver}_{\mathsf{pk}}$ excludes such perturbations.

**Theorem E.2** (Unforgeable watermarking)**.** *Let $Q$ be any language model, and let $\Phi$ be an efficiently computable predicate. Let $\gamma \in [0, 1/4)$ be a constant, and let $\mathsf{PRC}[\mathsf{Gen}, \mathsf{Enc}, \mathsf{Dec}]$ be an ideal PRC with respect to the composite predicate $\Phi \circ \mathsf{Ham}_{\gamma n}$, with codeword size $n$ and message size $k$. Let $\mathsf{DS}[\mathsf{Gen}, \mathsf{Sign}, \mathsf{Ver}]$ be a digital signature scheme with signature size $k$.*

*Then, for any constant $\beta \geq 0$ satisfying $\beta < \gamma$, or $\beta = \gamma = 0$, the above watermarking scheme $(\mathsf{Gen}, \mathsf{Wat}, \mathsf{Ver})$ is* undetectable *and satisfies the following faithfulness guarantees.*

**Unforgeability:** *For any white-box PPT adversary $\mathcal{A}$ that outputs a string in $\{0,1\}^{2n}$,*

$$\Pr\left[ \begin{array}{ll} (\mathsf{pk}, \mathsf{sk}) & \leftarrow \mathsf{Gen}(1^\lambda) \\ \zeta & \leftarrow \mathcal{A}^{\mathsf{Wat}_{\mathsf{sk}}, \mathsf{Ver}_{\mathsf{pk}}}(1^\lambda, \mathsf{pk}) \end{array} : \mathsf{Ver}(\mathsf{pk}, \zeta) > \mathcal{S}_s(\zeta) \right] \leq \mathrm{negl}(\lambda) \,,$$

*where $s \in \mathcal{T}$ denotes the time at which $\zeta$ is output (the adversary may continue interacting after this point), and $\mathcal{S}_s$ is the ideal attribution function defined by Eq. (4), with $\mathcal{R}_s$ defined by the modified $\beta$-potential selection rule in Eq. (6).*

**Robustness:** *For any black-box PPT adversary $\mathcal{A}$ that outputs a string in $\{0,1\}^{2n}$,*

$$\Pr\left[ \begin{array}{ll} (\mathsf{pk}, \mathsf{sk}) & \leftarrow \mathsf{Gen}(1^\lambda) \\ \zeta & \leftarrow \mathcal{A}^{\mathsf{Wat}_{\mathsf{sk}}, \mathsf{Ver}_{\mathsf{pk}}}(1^\lambda) \end{array} : \mathsf{Ver}(\mathsf{pk}, \zeta) < \mathcal{R}_s^\varphi(\zeta) \right] \leq \mathrm{negl}(\lambda) \,,$$

*where $s \in \mathcal{T}$ denotes the time at which $\zeta$ is output (the adversary may continue interacting after this point), $\varphi$ is the modified predicate defined in Eq. (5), and $\mathcal{R}_s$ is the ideal attribution function induced by the modified $\beta$-potential selection rule in Eq. (6).*

## F. Digital Signatures

For completeness, we recall the definition of digital signature schemes. Digital signature schemes allow a signer to associate a message with its publicly-verifiable signature. Parties without the signing key cannot forge valid signatures on messages

the signer has not signed. For clarity, we define them with respect to a fixed message size $n$. If no message size is specified, we assume the message space is $\{0, 1\}^*$, since longer messages can be handled by first applying a collision-resistant hash function to compress them into $n$ bits before signing.

**Definition F.1** (Digital signature scheme). *A digital signature scheme with message size $n = n(\lambda)$ and signature size $k = k(\lambda)$ is a triple* $(\mathsf{Gen}, \mathsf{Sign}, \mathsf{Ver})$ *of PPT algorithms such that for all $\lambda \in \mathbb{N}$,*

- $\mathsf{Gen}(1^\lambda)$ : *generates a pair of public and secret keys* $(\mathsf{pk}, \mathsf{sk})$.

- $\mathsf{Sign}(\mathsf{sk}, y)$ : *takes in a message $y \in \{0, 1\}^n$, and outputs a signature $\sigma \in \{0, 1\}^k$.*

- $\mathsf{Ver}(\mathsf{pk}, y, \sigma)$ : *takes in $y \in \{0, 1\}^n$, $\sigma \in \{0, 1\}^k$, and outputs a binary decision.*

**Correctness:** *For any $y \in \{0, 1\}^n$,*

$$\Pr\left[ \begin{array}{rl} (\mathsf{pk}, \mathsf{sk}) & \leftarrow \mathsf{Gen}(1^\lambda) \\ \sigma & \leftarrow \mathsf{Sign}(\mathsf{sk}, y) \end{array} \ : \ \mathsf{Ver}(\mathsf{pk}, y, \sigma) = 1 \right] = 1 \, ,$$

**Unforgeability:** *For any PPT adversary $\mathcal{A}$,*

$$\Pr\left[ \begin{array}{rl} (\mathsf{pk}, \mathsf{sk}) & \leftarrow \mathsf{Gen}(1^\lambda) \\ (y, \sigma) & \leftarrow \mathcal{A}^{\mathsf{Sign}_{\mathsf{sk}}(\cdot)}(1^\lambda, \mathsf{pk}) \end{array} \ : \ \big(\mathsf{Ver}(\mathsf{pk}, y, \sigma) = 1\big) \ \wedge \ \big(y \notin \Pi\big) \right] \leq \mathrm{negl}(\lambda) \, ,$$

*where $\Pi$ is the ledger of the interaction between $\mathcal{A}$ and the signing oracle $\mathsf{Sign}_{\mathsf{sk}}(\cdot)$ containing only message blocks.*

